# Structural basis of the promiscuity of the unusual Fe(II) and 2-oxoglutarate dependent human aspartate/asparagine-β-hydroxylase

Protein-hydroxylation catalysed by Fe(II) and 2-oxoglutarate (2OG) dependent oxygenases is an important regulatory mechanism in human biology. Such oxygenases typically coordinate their Fe(II) cofactor via a conserved triad of an aspartate- or glutamate- and two histidine-residues. By contrast, aspartate/ asparagine β-hydroxylase (AspH), which catalyses asparagine/aspartate-residue oxidation in epidermal growth factor-like domains (EGFDs), has only two histidine-residues (H679, H725), with a water occupying the site normally occupied by an aspartate- or glutamate-residue. We describe mechanistic studies with catalytically active AspH crystals. Turnover studies with single crystals under cryogenic conditions give (3$R$)-hydroxylated EGFDs with the product alcohol coordinating Fe(II) *trans* to H725. Time-resolved serial crystallography of microcrystals using an acoustic droplet ejection system, coupled to X-ray emission analyses, demonstrate turnover within 1.5 s, giving a product complex in which Fe(II) is regenerated. Solution and crystallographic studies with the $O_2$ surrogate nitric oxide imply $O_2$ binds to Fe(II) *trans* to H725. The additional Fe-chelating water is maintained throughout AspH catalysis and is not directly involved in substrate hydroxylation, because $O_2$ is the sole oxygen source in alcohol products, as shown by $^{18}O$ labelling studies. The results reveal how AspH accommodates both aspartate- and asparagine-substrates and will assist in efforts targeting AspH for cancer treatment.

2-Oxoglutarate (2OG)- and Fe(II)-dependent oxygenases catalyse a range of oxidative reactions, including hydroxylations, halogenations, demethylations, and oxidative ring closures[1]. In humans, they have roles in transcriptional regulation, lipid metabolism, epigenetics, collagen biosynthesis, and hypoxia sensing[2–5]. Inhibitors of the 2OG-dependent hypoxia-inducible factor-α (HIF-α) prolyl hydroxylases (PHDs) and γ-butyrobetaine hydroxylase (BBOX) are clinically used[6–8]. 2OG oxygenases other than the PHDs and BBOX are current medicinal chemistry targets for cancer treatment, including JmjC histone demethylases[9] and aspartate/asparagine β-hydroxylase (AspH)[10–14].

AspH is a 2OG- and Fe(II)-dependent oxygenase that catalyses the C3 hydroxylation of aspartate- and asparagine-residues in multiple epidermal growth factor-like domains (EGFDs) located in the endoplasmic reticulum (ER)[15,16]. EGFDs are compact ~30–50 residue domains that typically contain three intradomain disulfide bonds[17]. Recent studies have revealed that AspH only accepts EGFDs as substrates when they have a non-canonical disulfide bond connectivity (C1-C2, C3-C4, C5-C6), rather than the canonical C1-C3, C2-C4, C5-C6 disulfide connectivity that is observed in most EGFD structures, suggesting a role for AspH in regulating thiol/disulfide biochemistry in the ER[18]. The results of crystallographic studies on AspH complexes using Mn(II) as a catalytically inactive Fe(II) surrogate and a derivative of the EGFD1 of human coagulation factor X (hFX) with the non-canonical C3-C4 disulfide (hFX-Asp) show that substrate binding not only involves the AspH 2OG oxygenase domain, but also an adjacent tetratricopeptide repeat (TPR) domain located N-terminal to the oxygenase

e-mail: lennart.brewitz@ichf.edu.pl; jfkern@lbl.gov; christopher.schofield@chem.ox.ac.uk; patrick.rabe@chem.ox.ac.uk

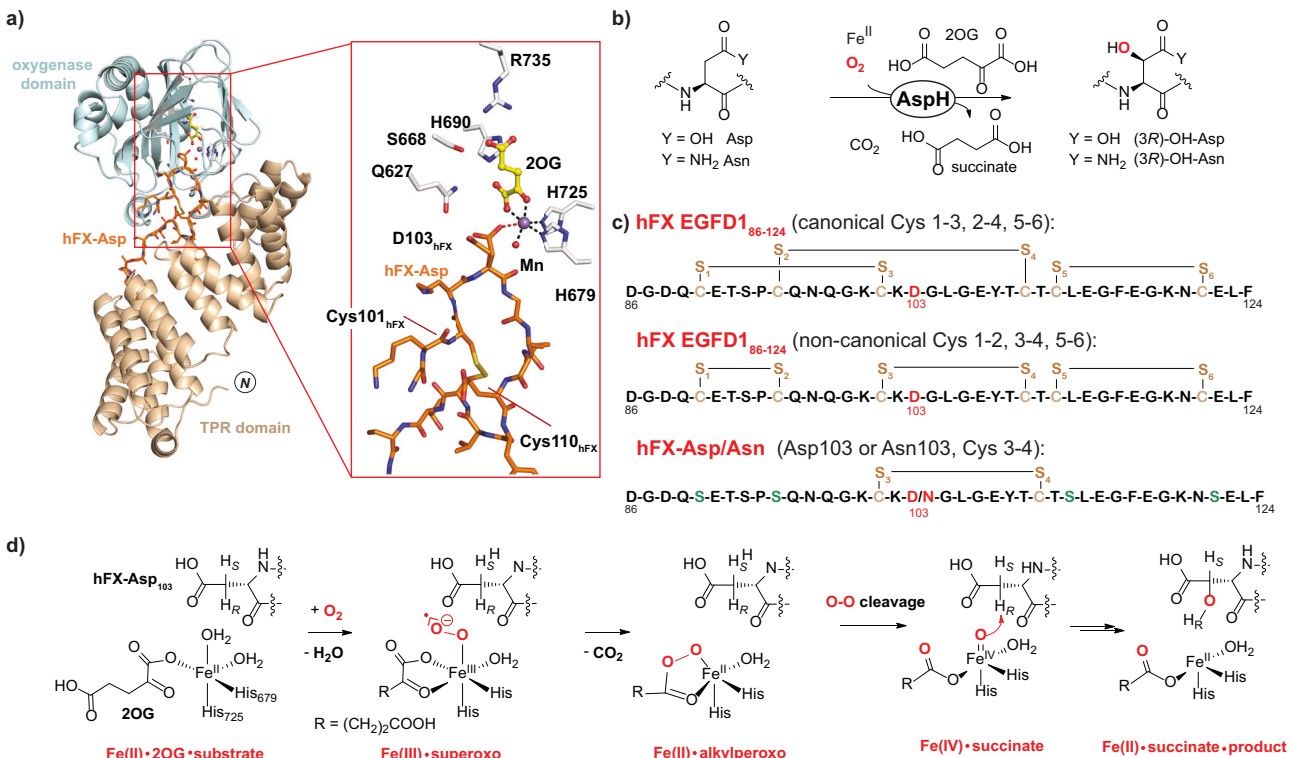

**Fig. 1 | Structure and proposed mechanism of AspH. a** View from a reported AspH:Mn:2OG:hFX-Asp complex crystal structure (PDB ID: 8RE9) showing the oxygenase domain (cyan), the TPR domain (wheat) and the hFX-Asp substrate (orange)[40]. The crystal structure reveals metal-coordinating and 2OG-binding residues: H679 and H725 (for Mn) and S668, H690 and R736 (for 2OG). Note that Asp103$_{hFX}$ is partially oriented to interact with the Q627 sidechain (likely the productive conformation enabling O$_2$ binding) and partially to coordinate Mn (a likely non-productive conformation preventing O$_2$ binding). **b** AspH-catalysed hydroxylation of Asp/Asn-residues in EGFDs. **c** The canonical (Cys 1–3, 2–4, 5–6; top) and non-canonical (Cys 1–2, 3–4, 5–6; middle) EGFD disulfide patterns; the synthetic hFX–Asp/Asn (Cys 3–4, bottom) substrate[18] used in this work. The AspH-hydroxylation site (Asp103$_{hFX}$ and Asn103$_{hFX}$) is in red, cysteine sulphurs are in wheat; and substituted serine residues are in green. **d** Proposed key intermediates during AspH catalysis (see also Supplementary Fig. 1)[39]. hFX = human coagulation factor X, TPR domain = tetratricopeptide repeat domain.

domain (Fig. 1)[18,19]. The ability of AspH to catalyse hydroxylation of multiple EGFD substrates resembles that of several other human 2OG oxygenases, including factor inhibiting HIF-α (FIH) and Jumonji-C domain-containing protein 6 (JMJD6)[20,21].

Many 2OG oxygenases appear to employ a largely conserved catalytic cycle, wherein coordination of 2OG to Fe(II) is normally followed by substrate binding at the active site[22]. O$_2$ binding to Fe(II) then generates a Fe(III):2OG:superoxide intermediate that undergoes oxidative decarboxylation to yield CO$_2$ and, at least in one case, a succinyl peroxide intermediate[23–25]. The latter fragments to produce an Fe(IV) = O (ferryl) species[26–31], which abstracts a hydrogen from the substrate[32]; radical rebound reaction with Fe(III)–OH forms the hydroxylated product and regenerates Fe(II) (Fig. 1, Supplementary Fig. 1)[32]. Variations on the consensus mechanism can occur, including with respect to the coordination sites to which O$_2$ and the 2OG C1 carboxylate bind[33,34].

Unlike most human 2OG oxygenases which typically coordinate Fe(II) via a conserved HXD/E…H triad of residues[1,35], the Fe(II) in AspH is coordinated by two histidine-residues (H679 and H725), with a water molecule (W1) occupying the normal aspartate-/glutamate-carboxylate site[18]. This AspH Fe(II)-binding motif resembles that of the 2OG-dependent halogenases (Supplementary Fig. 2)[36,37], suggesting that AspH may employ an atypical mechanism, although it apparently does not catalyse halogenation of EGFDs[18]. Computational and biochemical studies indicate the importance of second coordination sphere residues in stabilizing the water (W1) coordinating the Fe(II) of AspH [38,39].

The unusual Fe(II) coordination chemistry of AspH raises the question as to what extent it follows the consensus mechanism for

2OG dependent hydroxylases[1,39]. In particular, it is unknown to which Fe(II) coordination site O$_2$ binds to and in which position the Fe(IV) = O intermediate, which is usually responsible for hydrogen abstraction and subsequent hydroxylation, is formed.

A recently reported catalytically inactive AspH:Mn:2OG:hFX-Asp complex crystal structure revealed two conformations of Asp103$_{hFX}$ (i.e., the hydroxylation site of hFX-Asp): one in which it interacts with the Q627 sidechain, leaving the coordination site *trans* to H725 vacant for potential O$_2$ binding, and one in which the Asp103$_{hFX}$ carboxylate coordinates directly to Mn, potentially blocking O$_2$ access (Fig.1)[40]. Since the substitution of Fe(II) for inactive Mn(II) inhibits catalysis[41], investigating the native AspH:Fe(II) complex is important to determine whether this non-productive Asp103$_{hFX}$ conformation is relevant to Fe-dependent catalysis.

Time-resolved serial femtosecond crystallography (SFX) studies using an X-ray free electron laser (XFEL) enable mechanistic investigations on enzymes at atomic resolution in real time at ambient temperature[42–44], including on redox-labile intermediates which are difficult to analyse using traditional synchrotron radiation due to reactions induced by X-ray photoelectric effects[45,46]. By integrating acoustic droplet ejection (ADE)-type sample delivery with femtosecond crystallography[47,48], experimental throughput and temporal precision have been significantly enhanced, enabling precise initiation of reaction and synchronized capture of intermediate states. The combination of SFX with complementary techniques, such as X-ray emission spectroscopy (XES), enables time-resolved analysis of oxidation states, adding a distinct dimension to the mechanistic understanding of catalytic cycles. These approaches have provided insight

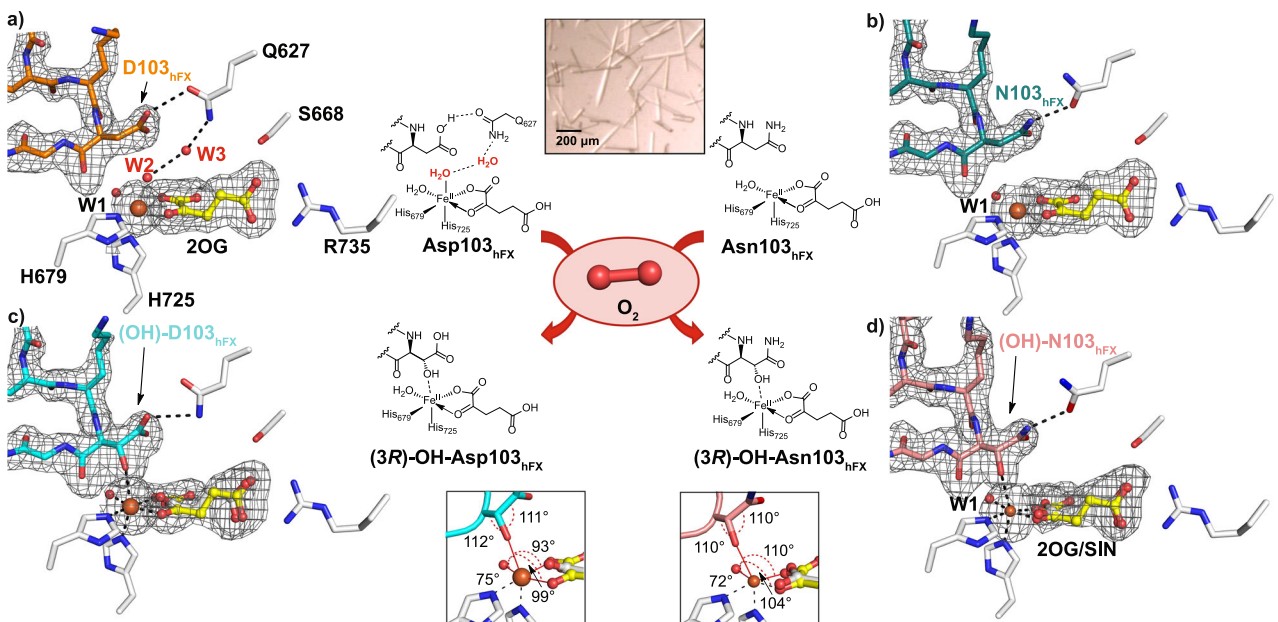

**Fig. 2 | Reaction of O₂ with anaerobic AspH:Fe:2OG:hFX-Asp/Asn complexes.** Active site view and Polder omit maps (3.0 σ contour level) for: **a** AspH:Fe:2OG:hFX-Asp (orange, PDB: 9FVZ), **b** AspH:Fe:2OG:hFX-Asn (teal, PDB: 9FVX), **c** AspH:Fe:2OG/succinate:hFX-(OH)Asp (cyan, PDB: 9FVY), and **d** AspH:Fe:2OG/succinate:hFX-(OH)Asn (pink, PDB: 9FVW) reveal catalysis *in crystallo*. The insets show key angles of the apparently slightly distorted tetrahedral geometry of the Asn103$_{hFX}$ and Asp103$_{hFX}$ hydroxyl groups and a distorted octahedral Fe coordination geometry in the AspH:product complexes. Residues involved in Fe (H679, H725) and 2OG interactions (S668, R735) are in sticks, Fe: orange sphere; 2OG and succinate: yellow; coagulation factor X derived hFX-Asp/Asn substrates: different colours.

into the mechanisms and structures of metalloenzymes[49–51], including 2OG oxygenases[25,52].

Here, we report investigations on the AspH mechanism, using both conventional cryogenic X-ray crystallography of catalytically active single crystals and room temperature substrate and product complex structures, obtained using SFX at an XFEL source. The combined crystallographic results provide information on conformational changes involving a hydrogen bonding network at the AspH active site during catalysis. Solution-phase studies, including EPR spectroscopy and nitric oxide (NO) binding experiments, together with isotopic labelling and X-ray emission spectroscopy (XES) analyses reveal how AspH accommodates both aspartate- and asparagine-substrates for stereoselective hydroxylation and provide insight into the activation of O₂ and substrate hydroxylation within the atypical AspH active site.

## Results

### Crystallographic analysis of catalytically active AspH complexes

Reported AspH complex crystal structures have employed catalytically inert Mn(II) or Ni(II) as Fe(II) substitutes (Fig. 1a)[18,19], a difference that may perturb the fold and/or active site metal coordination geometry. We thus worked to obtain crystal structures of catalytically active AspH complexed with Fe(II). We obtained >500 mg of homogenous recombinant untagged AspH$_{315-758}$ in a single preparation following our recently reported optimized protocol[53]. This material was used to optimise AspH crystallization under anaerobic conditions with Fe(II), 2OG, and hFX-Asp, yielding needle-shaped crystals (precipitation buffer: 0.1 M bis-tris propane, pH 7.5, 0.2 M NaBr, 18% v/v PEG 3350). Data collection at Diamond Light Source (DLS) under cryogenic conditions and subsequent elucidation of the AspH structure revealed the presence of Fe, 2OG, and hFX-Asp (AspH:Fe:2OG:hFX-Asp; PDB: 9FVZ, 1.95 Å resolution). The overall fold in the AspH:Fe:2OG:hFX-Asp complex structure is similar to that observed in the corresponding reported aerobic AspH:Mn:2OG:hFX-Asp complex structure (PDB: 8RE9, RMSD = 0.156 Å, Supplementary Fig. 3)[40].

The Fe-ion in the AspH:Fe:2OG:hFX-Asp complex structure is coordinated via the H679 and H725 sidechains (2.3 and 2.1 Å, respectively), the 2OG C1 carboxylate and C2 carbonyl (2.1 and 2.3 Å, respectively), and two H₂O molecules: W1 located *trans* to the 2OG C2 carbonyl (2.1 Å), i.e. the site typically occupied by an aspartate/glutamate carboxylate in 2OG oxygenases[1,35], and W2 located *trans* to H725 (2.2 Å) (Fig. 2a). 2OG binding is stabilized via hydrogen bonding and electrostatic interactions with the sidechains of S668, H690 and R735, as observed in the corresponding AspH:Mn:2OG:hFX-Asp complex structure (Supplementary Fig. 3c)[19,40]. W1 binding is stabilised by hydrogen bonding with the sidechain carboxylate of the second coordination sphere D721 and the backbone carbonyl of the substrate Asp103$_{hFX}$ (2.9 and 2.6 Å, respectively; Fig. 2a).

In the AspH:Fe:2OG:hFX-Asp structure, modelling of W2 (the water located *trans* to H725) indicated partial occupancy (~50%) (Fig. 2a, Supplementary Fig. 3d+e). In the refined model, W2 interacts with a nearby water (W3; 3.1 Å), which interacts with the Q627 sidechain amide (2.6 Å) and the Asp103$_{hFX}$ sidechain carboxylate (2.9 Å). These interactions form a Fe-W2-W3-Q627-Asp103$_{hFX}$ hydrogen bond network (Supplementary Fig. 3d+f). Importantly, the Fe-coordination site *trans* to H725 is only partially occupied in the Fe-bound structure, whereas in the reported AspH:Mn:ligand:hFX-Asp complexes[18,19,40], it is filled, either by a water molecule or the substrate Asp103$_{hFX}$ carboxylate. In the AspH:Fe:2OG:hFX-Asp complex structure, no evidence for coordination of the Asp103$_{hFX}$ sidechain to the metal was observed (Fig. 2a). These differences suggest that the Fe(II) centre possesses greater coordination flexibility than the analogous Mn(II) complex. The observed six-coordinate, slightly distorted octahedral geometry with a weakly bound W2 in the Fe-bound complex supports the proposal that the W2 occupying coordination site becomes available for O₂ binding, consistent with reported computational predictions[39].

The corresponding AspH:Fe:2OG:hFX-Asn structure (PDB 9FVX, 1.90 Å resolution, Fig. 2b, Supplementary Fig. 3a+b) was obtained using identical conditions with the hFX-Asn substrate to inform on the effect of substituting Asp103$_{hFX}$ for Asn103$_{hFX}$. The overall fold (RMSD

0.152 Å) and active site geometry resemble those of the hFX-Asp complex (Supplementary Fig. 3f). Notably, however, only weak positive electron density (< 20%; Supplementary Fig. 3e) was observed for an Fe-bound water molecule located *trans* to H725 (corresponding to W2), and this water was therefore not refined (Fig. 2b, Supplementary Fig. 3).

The combined results imply that W2 is relatively weakly bound to Fe, suggesting that binding of the EGFD substrate may influence W2 binding and dissociation, thereby potentially modulating $O_2$ binding and catalytic efficiency. Such a mechanism may help explain the wide variation in the observed extent of EGFD hydroxylation (0-100%) even when the preferred substrate consensus sequence for AspH is present[54–56].

The absence of W2 in the hFX-Asn complex affects the conformation of the Q627 sidechain, which adopts a conformation enabling a stronger interaction with the Asn103$_{hFX}$ sidechain amide (2.8 Å) than with the Asp103$_{hFX}$ carboxylate (3.0 Å) in the corresponding hFX-Asp structure. W3 is absent in the hFX-Asn complex structure, thus the Fe–W2–W3–Q627–Asp103$_{hFX}$ hydrogen-bond network is not formed (Supplementary Fig. 3e+f). These observations suggest that the intact hydrogen bond network is not essential for hydroxylation, but may contribute to tuning substrate selectivity and reactivity. Notably, no evidence was obtained for direct Fe-coordination by the Asn103$_{hFX}$ carboxamide, contrasting with the AspH:Mn:2OG:hFX-Asp structures where the Asp103$_{hFX}$ carboxylate was modelled to coordinate the Mn with 50% occupancy (Supplementary Fig. 3i)[19,40]. Collectively, the differences between the Fe and Mn structures highlight the importance of employing catalytically-relevant Fe(II) when investigating the coordination chemistry of 2OG oxygenases and, by implication, related metalloenzymes.

To investigate the effects of the hFX-Asp to hFX-Asn substitution on AspH catalysis, we performed solution-based kinetic studies with hFX-Asn using solid phase extraction coupled to mass spectrometry (SPE-MS) assays, comparing the results to those with hFX-Asp[57]. The results reveal that the AspH $k_{cat}/K_m$ value for Fe(II) with hFX-Asn was similar, within experimental error, to that reported with hFX-Asp[57]. For 2OG, the $k_{cat}/K_m$ for hFX-Asn was ~4-fold lower than that reported for hFX-Asp[57]. Interestingly, direct competition studies of hFX-Asn and hFX-Asp reveal that AspH catalysed hFX-Asp oxidation occurs more efficiently than with hFX-Asn (Supplementary Fig. 4), suggesting that the Fe-W2-W3-Q627-Asp103$_{hFX}$ hydrogen bond network supports catalytic efficiency. These results imply that turnover of Asn-containing EGFDs is less efficient than Asp-containing EGFDs, under equivalent conditions. Although the biological significance of this difference remains to be established, it raises the possibility that substrate-dependent variations in 2OG affinity may contribute to the variable extents of EGFD hydroxylation observed in cells. Future work is required to determine whether such kinetic differences play a regulatory role in AspH activity in vivo.

## Investigations on the $O_2$ binding site of AspH
Anaerobic AspH:Fe:2OG:hFX-Asp single crystals were exposed to nitric oxide (NO), a catalytically inactive $O_2$ mimic, to investigate whether $O_2$ binds to the Fe(II) coordination site partially occupied by W2 in the AspH:Fe:2OG:hFX-Asp structure. Incubation of crystals in NO saturated precipitation solution resulted in an AspH:Fe:2OG:NO:hFX-Asp structure (PDB: 9HO3, 2.39 Å resolution; Fig. 3a, Supplementary Fig. 5a). NO displaces W2 to coordinate to Fe(II) *trans* to H725 (2.0 Å) with the oxygen of NO 3.0 Å from the α-carbon of Asp103$_{hFX}$; note that electron density in the position of W3 was not observed. To test for the presence of NO at the active site, modelling of $H_2O$ or a product alcohol [(3 R)-OH-Asp103$_{hFX}$] was attempted: neither model reflected the observed electron density (Supplementary Fig. 5c).

EPR was then employed to investigate NO binding to AspH in solution. Samples were prepared by exposing a solution of the anaerobic AspH:Fe:2OG:hFX-Asp complex to 0.1% NO in $N_2$. The results show that NO binds to form a single characteristic high-spin (*hs*) {FeNO}[7] $S = 3/2$ state, simulated with effective g-values of $g_1 = 4.15$, $g_2 = 3.915$, and $g_3 = 2.0017$ (Fig. 3b, Supplementary Fig. 6)[58]. Without hFX-Asp, evidence for a mixture of nitrosyl iron centre species was observed, in particular dinitrosyl iron complexes (DNIC, Supplementary Fig. 7), indicating that NO occupies more than one Fe(II) coordination site, possibly in a concentration-dependent manner. UV-Vis analyses of the same samples further supports binding of NO to the AspH:Fe:2OG:hFX-Asp complex, as shown by increased absorption in the 300–400 nm region, an observation more pronounced without hFX-Asp (Supplementary Fig. 8). This absorption band is characteristic of metal–nitrosyl charge-transfer transitions[59], reflecting Fe–NO bond formation and changes in the Fe(II) coordination environment.

Further evidence for an AspH:Fe:2OG:NO:hFX-Asp complex was obtained by the observation that irradiation with unfiltered white light resulted in apparent NO dissociation from the complex, as implied by the disappearance of the *hs* {FeNO}[7] signal at temperatures of 10 K, 15 K, 25 K, and 30 K, which was recovered upon removal of the light source and/or increasing of the temperature to 77 K (Supplementary Fig. 9). The photolysis-recovery EPR experiments imply an activation energy (Ea) for re-binding of NO of ~0.12 kJ/mol (Supplementary Figs. 10–11). This low activation barrier value is substantially lower than those reported for analogous studies with other metalloenzymes; for instance, with cytochrome c oxidase, Ea values for NO re-binding measured at 50 K are an order of magnitude higher, with reported ranges between 1–5 kJ/mol[60,61]. These comparisons imply that the {FeNO}[7] species in AspH exists in a labile coordination state, supporting its role as a mimic of a transient Fe–$O_2$ complex. The combined results support the likely site of $O_2$ binding as the Fe(II) coordination site *trans* to H725, i.e., the site partially occupied by W2 in the AspH:Fe:2OG:hFX-Asp complex structure (Fig. 2a), a configuration that mirrors the proposed $O_2$ binding modes observed in some other 2OG oxygenases, including TauD and PHD2[62–64].

We assessed the binding of NO to the microcrystalline sample of AspH:Fe:2OG:hFX-Asp and AspH:Fe:2OG:hFX-Asn by EPR spectroscopy, using $FeSO_4$ as an Fe(II) source. In the microcrystalline state, NO did not appear to coordinate to the iron after exposure to 0.1% NO in $N_2$. Following exposure to an NO saturated precipitation solution, however, the microcrystalline samples resulted in formation of the *hs* {FeNO}[7] $S = 3/2$ spin state (Fig. 3b, trace iia+b, Supplementary Table 1). Overall, in frozen solution, the rhombicity of the {FeNO}[7] complex was approximately twice that of the microcrystalline state, indicating a more symmetric ligand field in the latter which may influence the energetic landscape of the catalytic intermediate. Interestingly, an anomalous NO species at g ~ 2 was also observed in microcrystals, something reminiscent of observations with other non-heme iron systems[58].

## Binding of isothiocyanate to AspH
We investigated the ability of (pseudo)halides to bind with AspH using X-ray crystallography to inform on possible reasons why AspH does not catalyse EGFD halogenation despite sharing a similar Fe(II) coordination geometry with 2OG-dependent halogenases[18,65]. Crystals grown in the presence of potassium thiocyanate (KSCN) revealed electron density for AspH, Fe, 2OG, and hFX-Asn (PDB: 9FVU, resolution 1.70 Å). Notably, additional positive electron density was observed near the Fe W2/$O_2$ binding site *trans* to H725: The electron density was modelled with isothiocyanate (70% occupancy) coordinating to Fe via its nitrogen (M–N = C = S) with an Fe–N–C angle of ~137° (AspH:Fe:2OG:hFX-Asn:NCS complex; Fig. 3c+d, Supplementary Fig. 12a+b), consistent with reported isothiocyanate complexes[66]. We also obtained the corresponding AspH:Mn:2OG:hFX-Asn:NCS complex, which showed a similar AspH fold as in the corresponding Fe

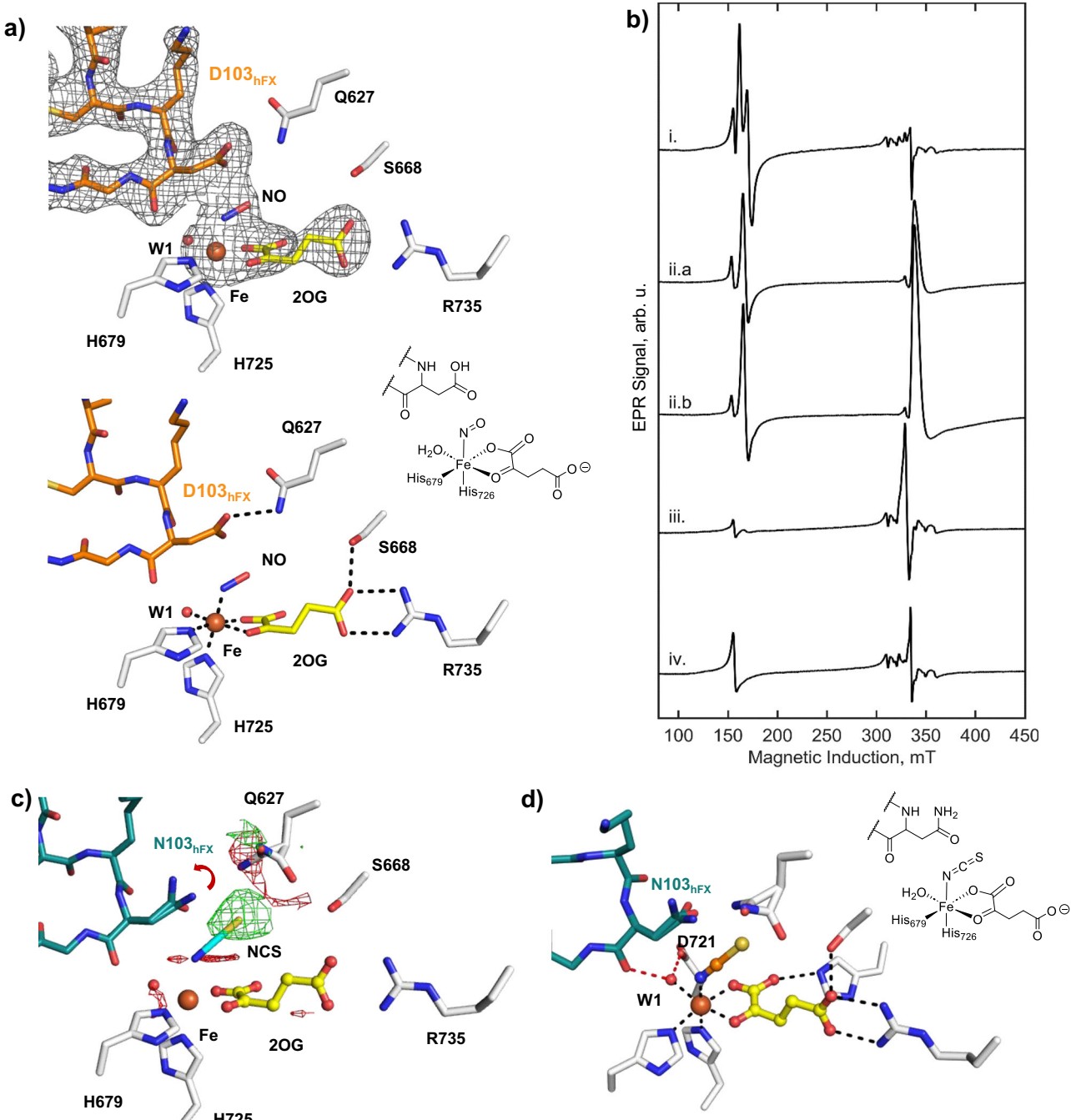

**Fig. 3 | NO binding studies inform on the possible AspH O₂ binding site. a** Active site view/Polder omit map carved around hFX-Asp, Fe, 2OG, and NO (3.0 σ contour level) of the AspH:Fe:2OG:NO:hFX-Asp structure (PDB: 9HO3). Key interactions: black dashed lines; Fe: orange sphere; 2OG: yellow; hFX-Asp: orange. **b** X-band EPR at 5 K of AspH:Fe:2OG samples at 100 μW microwave power, 9.3839(3) GHz microwave frequency, and 1 mT field modulation: (i) 1 mM solution of AspH:Fe:2OG, NO and hFX-Asp; (ii.a) microcrystalline slurry of AspH:Fe:2OG, NO and hFX-Asp; (ii.b) microcrystalline slurry of AspH:Fe:2OG, NO and with hFX-Asn; (iii.) AspH:Fe:2OG and NO; (iv) AspH:Fe:2OG and hFX-Asp. Data for an anaerobic structure (RMSD = 0.135 Å), but with full isothiocyanate occupancy

buffer blank were subtracted, arb. units = arbitrary units. **c** $F_{obs} - F_{obs}$ isomorphous difference map of the anaerobic AspH:Fe:2OG:NSC:hFX-Asn complex (PDB: 9FVU), compared to the NO complex (PDB: 9HO3). A red arrow indicates the rearranged conformation of the substrate Asn103hFX-carboxamide. **d** Key interactions of the Fe and 2OG coordinating residues in the AspH:Fe:2OG:hFX-Asn:NCS (PDB: 9FVU) complex. Binding of the Fe-bound H₂O (W1) is stabilised by second sphere residues D721 and backbone carbonyl of Asn103hFX (red dashed lines, Supplementary Fig. 12).

(PDB: 9FVV, Supplementary Fig. 12c+d).

Superimposition of the hFX-Asn complex structures, either with or without isothiocyanate, reveals that binding of isothiocyanate induces local conformational changes. In the presence of isothiocyanate, the Q627 sidechain amide reorients away from the

carboxamide of Asn103hFX, increasing the separation beyond hydrogen bonding distance (>5 Å). Q627 adopts a conformation that enables a potential hydrogen bond with the sidechain hydroxyl of S688 (2.9 Å), which forms a hydrogen bond with the 2OG C5 carboxylate (2.7 Å) (Supplementary Fig. 12e,f). Interestingly, while isothiocyanate coordination was observed in the AspH:hFX-Asn

structures, it was not evident when substituting hFX-Asn for hFX-Asp. This observation may reflect a stronger interaction between the Q627 sidechain and the Asp103$_{hFX}$ sidechain carboxylate than that with the Asn103$_{hFX}$ sidechain amide, as supported by the observed substrate preference of AspH for hFX-Asp over hFX-Asn (Supplementary Fig. 4).

The observation that a pseudohalogen ion can bind to AspH-complexed Fe(II) is of interest, because, although isothiocyanate binds in proximity of the EGFD substrate, turnover studies with AspH in the presence of excess thiocyanate/other pseudohalogens did not reveal evidence for EGFD (pseudo)halogenation[18]. Although the AspH Fe(II) coordination geometry resembles that of 2OG dependent halogenases[36,37], isothiocyanate binding is observed at the W2/O$_2$ binding site rather than the W1 Fe(II) coordination site (*trans* to the 2OG C2 carbonyl), where halides typically bind in 2OG halogenases to enable substrate halogenation. The location of isothiocyanate at W2 therefore, represents a non-productive geometry, which likely impairs both O$_2$ activation and subsequent potential substrate (pseudo) halogenation.

## Crystallographic analysis of AspH product complexes

Exposure of AspH:Fe:2OG:hFX-Asp/hFX-Asn crystals to air for ≥20 min at room temperature gave product complex structures with (3*R*-OH)-Asp103$_{hFX}$ (PDB: 9FVY, 1.90 Å resolution) and (3*R*-OH)-Asn103$_{hFX}$ (PDB: 9FVW, 1.90 Å resolution), respectively, both refined with complete product occupancy (data collected under cryogenic conditions at the Diamond Light Source), implying single, O$_2$-dependent substrate turnover *in crystallo* (Fig. 2c+d, Supplementary Fig. 13). Both product-complex structures reveal near identical AspH folds (Fig. 2c+d, Supplementary Fig. 13c), very similar to those of the corresponding substrate complexes. A mixture of 2OG and succinate (each modelled with 50% occupancy based on best agreement with electron density and B-factors) was refined in both product structures, consistent with partial re-binding of 2OG to the active site after substrate hydroxylation, as 2OG was present in excess during crystallization. Because AspH performs only a single turnover *in crystallo*, the large hydroxylated EGFD product remains bound due to lattice and domain interactions, whilst the smaller 2OG cosubstrate and succinate coproduct can exchange with the surrounding mother liquor via solvent channels.

The cryogenic AspH:Fe:2OG/succinate:hFX-(3*R*-OH)-Asp/hFX-(3*R*-OH)-Asn product structures imply stereospecific oxidation to the (3 *R*)-OH-Asp103$_{hFX}$ and (3 *R*)-OH-Asn103$_{hFX}$ products (Fig. 2c+d), consistent with pioneering work on assigning the configuration of 3-OH-Asn/Asp in EGFDs[15,67]. We validated the product stereochemistry by refining the structures with the (3*S*)-OH-Asp103$_{hFX}$ and (3*S*)-OH-Asn103$_{hFX}$, resulting in models that did not reflect the observed electron density (Supplementary Fig. 13d+e).

The C3 hydroxy group of (3 *R*)-OH-Asp/Asn103$_{hFX}$ was observed to coordinate to Fe *trans* to H725 in both AspH:Fe:2OG/succinate:hFX-(3*R*-OH)-Asp/Asn product structures (2.5 Å; Fig. 2c+d), replacing the Fe-bound W2 observed in the AspH:Fe:2OG:hFX-Asp substrate structure and occupying the vacant Fe-coordination site in the AspH:Fe:2OG:hFX-Asn structure. Note the absence of W3 in both product structures (Fig. 2c+d). The orientations of AspH active site residues in both product complex structures are similar to those observed in the substrate complex structures with the notable exception of the Q627 sidechain, which adopts a conformation that likely enhances interactions with the sidechain carboxylate/amide of Asp/Asn103$_{hFX}$. The distorted octahedral Fe-coordination geometry observed in the product structures likely affects the electronic and steric properties of their Fe centres, possibly modulating the stability of the product complex and promoting product release in solution. The observed lack of product release *in crystallo* is likely a result of the interactions of the hydroxylated EGFD product with both the 2OG oxygenase and TPR domains in the crystalline lattice and the reduced conformational flexibility of AspH *in crystallo*; by contrast,

the smaller 2OG cosubstrate remains exchangeable on the experimental timescale; note that AspH binds its substrates via an induced fit mechanism associated with substantial conformational changes involving both the 2OG oxygenase and the TPR domains[18].

## Optimisation for serial crystallography

Given its ability to probe reactions in crystals with minimal radiation damage, we explored the potential of serial femtosecond crystallography (SFX) to capture AspH substrate and product complexes at physiological temperature and pressure. Crystallisation conditions were optimized to enable robust microcrystal slurry formation ( ~ 2 μm x 2 μm x 60 μm) of the AspH:Fe(II):2OG:hFX-Asp complex under strictly anaerobic conditions (4 °C, <2 ppm O$_2$). The first SFX data set was collected at the Macromolecular Femtosecond crystallography (MFX)[68] instrument of the Linac Coherent Light Source (LCLS) using the drop-on-demand sample delivery system[48] and demonstrated that well-ordered anaerobic AspH microcrystals manifest diffraction properties suitable for time-resolved analyses (PDB: 9FW0, 1.95 Å resolution; Fig. 4a, Supplementary Fig. 15a). The structure was very similar to its cryogenic analogue, including with respect to the Fe, 2OG and hFX-Asp binding modes (RMSD = 0.254 Å, Supplementary Fig. 16a). Whilst full occupancy of W2 was observed in the room temperature structure, W3 which was observed in the analogous cryogenic structure, was not detected. This change in the solvent network did not affect the conformation of Q627. These results confirmed the stability of the AspH complex under near physiologically relevant temperature and established a robust foundation for subsequent SFX and XES experiments.

Subsequent optimization, including washing of microcrystals to remove excess Fe(II), which may interfere with simultaneously performed XES analysis, yielded an improved structure (PDB: 9HO2, 1.83 Å resolution; Fig. 4b, Supplementary Fig. 15b), with a fold highly similar to that of the previously obtained room temperature structure (RMSD = 0.159 Å, Supplementary Fig. 16b). The apparent full occupancy of Fe(II) and 2OG confirmed that washing of the crystals with precipitant did not affect Fe(II)/2OG binding. However, the Fe-W2-W3-Q627-Asp103$_{hFX}$ hydrogen bonding network was partly restored in the improved room temperature hFX-Asp complex: Although both W2 and W3 were observed in full occupancy, the Q627 sidechain was observed in two conformations: one in which the sidechain amide is positioned to interact with the Asp103$_{hFX}$ carboxylate (3.0 Å) and W3 (2.9 Å) and one positioned to interact with the W625 (2.5 Å) and S668 (3.1 Å) sidechains.

## Monitoring AspH catalysis *in crystallo* using serial crystallography and emission spectroscopy

To monitor AspH catalysis over different time scales at physiologically relevant temperatures, complementary experimental procedures were employed: (i) rapid, millisecond–second O$_2$ exposure of microcrystals at the LCLS using the drop-on-demand system[48], enabling precise control of reaction time before diffraction ("in crystallo turnover" experiments); and (ii) slower, passive air exposure of crystallisation samples for 2–24 h at 4 °C, enabling observation of late-stage or equilibrium product states[69,70]. The controlled O$_2$ delivery at LCLS was chosen to capture early catalytic intermediates and single-turnover events that cannot be resolved through bulk air exposure.

We first characterised a relatively short-timescale reaction by exposing the microcrystal slurry to O$_2$ for 1.5 s using the drop-on-demand tape-drive sample-delivery system at LCLS. Regulation of the O$_2$ exposure time by altering the tape drive speed[48] yielded a room temperature AspH product structure with clear density for (3 *R*)-OH-Asp103$_{hFX}$ (PDB: 9HO1, 1.85 Å resolution; Fig. 4c, Supplementary Fig. 15c), indicating complete substrate turnover. The fold of this structure closely resembles that of the cryogenic structure (RMSD = 0.296 Å), with minor differences in the conformation of the

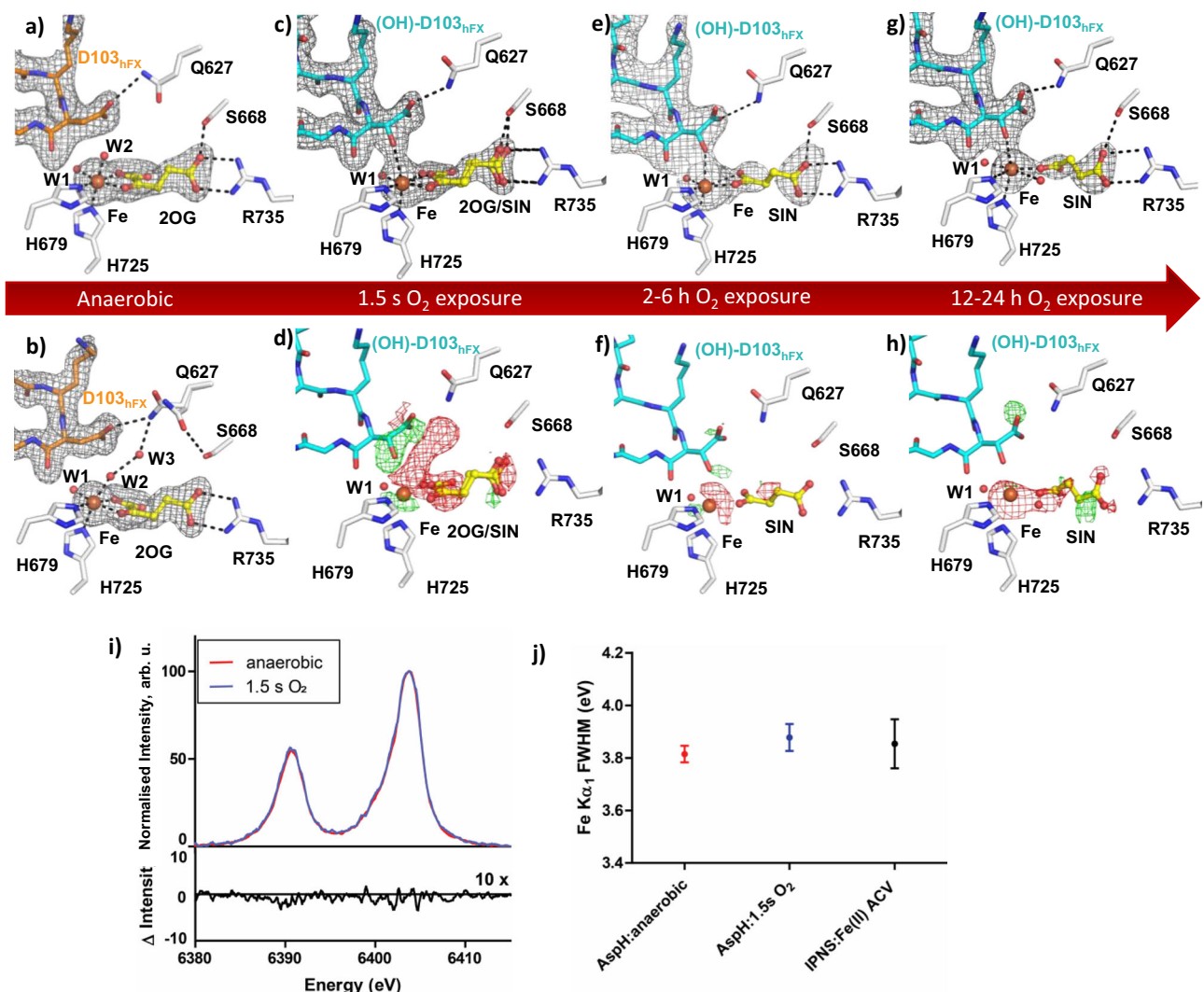

**Fig. 4 | Monitoring AspH turnover across increasing O₂ exposure times by serial crystallography and X-ray emission spectroscopy. a–c, e, g** Active site view/ Polder omit map (3.0 σ contour level) for room-temperature AspH:Fe:2OG:hFX-Asp complexes under anaerobic conditions and after 1.5 s, 2–6 h, and 12–24 h O₂ exposure with (**a**) anaerobic AspH:Fe:2OG:hFX-Asp (orange, PDB: 9FW0, SFX room temperature), **b** an improved anaerobic AspH:Fe:2OG:hFX-Asp (orange, PDB: 9HO2, SFX room temperature, after washing to remove excess Fe), **c** AspH:Fe:2OG/ succinate:hFX-(OH)Asp (cyan, PDB: 9HO1, SFX, room temperature) after 1.5 s O₂ exposure, **e** AspH:Fe:succinate:hFX-(OH)Asp (cyan, PDB: 9NHZ, after 2–6 h air exposure, SSX) and **g** AspH:Fe:succinate:hFX-(OH)Asp (cyan, PDB: 9NHZ, after 12–24 h air exposure, SFX). **d, f, h** F$_{obs}$ − F$_{obs}$ isomorphous difference map of (**d**) the 1.5 s O₂ exposed AspH:Fe:2OG/succinate:hFX-(OH)Asp complex relative to the anaerobic AspH:Fe:2OG:hFX-Asp complex, **f** AspH:Fe:succinate:hFX-(OH)Asp complex exposed to air (2–6 h) and **h** AspH:Fe:suiccinate:hFX-(OH)Asp complex

exposed to air (12–24 h) both relative to the AspH:Fe:2OG/succinate:hFX-(OH)Asp complex exposed to O₂ (1.5 s). **i** Fe Kα$_{1,2}$ XES of the anaerobic AspH:Fe:2OG:hFX-Asp complex before (red) and after 1.5 s O₂ exposure (blue, normalized to the peak maximum). A difference spectrum between the peak max normalized spectra for anaerobic and 1.5 s O₂ exposed samples (black, 10x enlarged) shows low variability between the two samples, supporting a complete turnover, arb. units = arbitrary units. (**j**) Full width at half maximum (FWHM) of the Fe Kα$_1$ XES peaks (in eV = electron volt) of the anaerobic AspH:Fe:2OG:hFX-Asp complex before (red) and after 1.5 s O₂ exposure (blue), and a comparison to that of a validated IPNS:- Fe(II):ACV complex[25]. The error bars were calculated as standard deviations of randomly sampled data sets created out of the total shots used in the construction of the spectra. The random sampling was performed for 50 iterations. For AspH:anaerobic and AspH:1.5 s O₂, total shots of 24426 and 10596 were used, respectively.

Q627 sidechain (Supplementary Fig. 16c). Only one conformation of the Q627 sidechain was apparent in which it is positioned to interact with the (3 R)-OH-Asp103$_{hFX}$ sidechain carboxylate (2.9 Å). By contrast, two conformations of the Q627 sidechain were observed in the corresponding anaerobic room temperature structure with the substrate (Supplementary Fig. 16c).

Isomorphous difference maps, which inform on local chemical transformations, were calculated between the anaerobic AspH:Fe(II):2OG:hFX-Asp room temperature complex and the product complex obtained after 1.5 s O₂ exposure, to visualize site-specific changes accompanying turnover (Fig. 4d,f,h). The isomorphous

difference maps reveal negative density for W2 and W3, indicating their absence in the O₂-exposed structure. Clear positive density was observed corresponding to a (3 R)-OH group on Asp103$_{hFX}$ (Fig. 4d). Negative density was also apparent around the 2OG C1 carboxylate, reflecting conversion of 2OG into succinate. 2OG and succinate could be modelled in a 1:1 ratio, an observation suggesting re-binding of 2OG after turnover, as observed for the cryo-product structures, suggesting that the smal 2OG cosubstrate can exchange with the surrounding mother liquor via solvent channels. Analysis of normalised B-factors in the anaerobic SFX structure and the O₂-exposed structure revealed high similarity, suggesting little dynamic variance in the AspH fold

following completion of the single turnover event (Supplementary Fig. 17).

Simultaneous collection of X-ray emission spectroscopy (XES) data on the anaerobic crystals and the 1.5 s $O_2$-exposed crystals informed on the Fe-oxidation state in the AspH complex before and after $O_2$ reaction (Fig. 4i+j, Supplementary Fig. 18). The XES analyses reveals little differences in the spectra of the anaerobic and the 1.5 s $O_2$-exposed crystals. This observation contrasts with observations for related systems, e.g. IPNS or sMMO, where a distinct feature in the XES difference spectrum was observed upon reaction of samples in the Fe(II) state when $O_2$ exposed[25,71], corresponding to a change from an Fe(II) to an Fe(III) state. The observed full width at half maximum (FWHM) results are indicative of an Fe(II) species in both the anaerobic AspH:Fe:2OG:hFX-Asp and the $O_2$-exposed AspH:Fe:2OG/succinate:(3R)-OH-Asp room temperature complexes based on comparison with data obtained with the anaerobic IPNS:Fe(II):substrate and $O_2$-exposed IPNS:Fe(III):substrate:superoxide complexes[25]. The slight broadening of the FWHM of ~50 meV upon $O_2$ mediated oxidation is likely not indicative of Fe-oxidation, as in that case a narrowing of the FWHM would be expected[25,71,72]. The broadening might be caused by a slight change in the ligand environment of the Fe site instead of a change in oxidation state. These results support the conclusion that, following a single catalytic turnover, AspH returns to the Fe(II) state, consistent with a mechanism involving Fe(II) regeneration (Fig. 4i+j, Supplementary Fig. 18).

We next examined longer equilibration time points by extending the $O_2$ exposure time, in order to investigate differences in the active site 2OG (present in excess under crystallisation conditions) to succinate ratios[18]. AspH:Fe:2OG:hFX-Asp microcrystals were exposed to air at 4 °C for ≥2 h with diffraction data collection using fixed target methods (both with serial synchrotron crystallography (SSX) at DLS and SFX at PAL-XFEL). Both the SSX (PDB: 9NHZ, 2.40 Å resolution; Fig. 4e+f, Supplementary Fig. 15d) and SFX (PDB: 9HO1, 2.14 Å resolution; Fig. 4g+h, Supplementary Fig. 15e) AspH structures could be refined with full occupancy for (3R)-OH-Asp103$_{hFX}$ and manifested highly similar overall folds (RMSD = 0.196 Å). Similar to the cryogenic product structure, replacement of the Fe-coordinating W2 by the product hydroxyl-group was observed along with a lack of W3. As observed in the 1.5 s $O_2$ exposed structure, B-factor analysis showed high overall similarity to the anaerobic structure (Supplementary Fig. 17).

Evidence for 2OG was not observed in the SSX and SFX structures obtained at longer time points (≥ 2 h $O_2$ exposure), contrasting with the structure obtained after 1.5 s $O_2$ exposure (Supplementary Fig. 16d); complete EGFD hydroxylation was observed for both 1.5 s and ≥ 2 h $O_2$ exposure. The presence of 2OG and succinate in the short time point may reflect (re)binding of 2OG subsequent to hydroxylation. At the longer time points, the amount of 2OG in solution may be depleted and that of succinate increased due to uncoupled oxidation of 2OG in solution catalysis arising from solvated AspH outside the crystal lattice or AspH at crystal surfaces, consistent with the observation of only succinate at the active site. Thus, while each AspH molecule within the crystal undergoes a single catalytic event, solution-phase turnover progressively may alter the small-molecule composition available for exchange with the crystal (Supplementary Fig. 19). Data for the SSX structure collected ~2-6 h post exposure to air reveal full occupancy of succinate, which was positioned to interact with Fe in a bidentate manner via a single carboxylate (Fig. 4e+f). By contrast, data for the SFX structure collected ~12–24 h post air exposure reveal ~0.85 occupancy of succinate (Fig. 4g+h), with the conformation of succinate differing from that observed in the SSX structure: succinate is positioned to coordinate the Fe in a monodentate mode, *trans* to H679. The Fe site *trans* to W1 featured additional density for which a $H_2O$ molecule was modelled (Supplementary Fig. 16e). Analysis of the electron density for the >2 h $O_2$ exposed

structure also indicates that the Fe-occupancy decreases upon exposure to air (Fig. 4f+h).

The combined results inform on a possible mechanism promoting succinate displacement from the active site following substrate hydroxylation, i.e., binding of the Fe-binding succinate carboxylate alters from a bidentate to a monodentate being mode, possibly correlating with strengthening of the interaction of the other succinate carboxylate with the R735 guanidinium group. The hydroxylated product and succinate can then dissociate enabling water and 2OG rebinding.

### Mass spectrometry studies of oxygen-atom incorporation

Given the potential for $O_2$ ligand exchange between Fe-coordinated waters (W1, W2) and the differing binding modes of Asp- and Asn-containing substrates, the source of the hydroxyl oxygen in the AspH products is of interest (Fig. 5). To investigate whether $O_2$ is the source of the O atom incorporated into the EGFD substrate, we incubated mixtures containing AspH, Fe(II), 2OG, LAA, and hFX-Asp or hFX-Asn with $^{18}O_2$ or $H_2{}^{18}O$ using a customised Schlenk-line setup[25] and analysed the samples by LC coupled to a QTOF mass spectrometer (Fig. 5, Supplementary Fig. 20). Controls without 2OG revealed no turnover with only the substrate mass being detected, *i.e.*, for hFX-Asp: 1047.2 Da $[M + 4H]^{4+}$. Incubation of the reaction mixture with $^{16}O_2$ gave the (3R)-$^{16}$OH-Asp103$_{hFX}$ product with a mass of 1051.2 Da $[M + 4H]^{4+}$, i.e., 4 mass shifts in the m/z = 4 charge state with respect to the substrate (Supplementary Figs. 20–21). Incubation of the mixture under $^{18}O_2$ gave the (3R)-$^{18}$OH-Asp103$_{hFX}$ product with a mass of 1051.7 Da $[M + 4H]^{4+}$, *i.e.*, with relative mass increments of 0.51 in the m/z = 4 charge state compared to the $^{16}O_2$ results (Fig. 5b); the 1051.2 Da $[M + 4H]^{4+}$ masses were not observed in the $^{18}O_2$ incubation mixtures. Analogous results were obtained when substituting hFX-Asp with hFX-Asn (Fig. 5c). Variations in the 2OG concentration (from 1 to 5 equiv.) and pH (7.0–8.5) did not affect the (3R)-OH-Asp103$_{hFX}$ product masses (Supplementary Fig. 22-23). Analysis of the mass shifts associated with the conversion of 2OG (146.03 Da) to succinate (120.04 Da; mass difference of 25.99 Da) in mixtures incubated with $^{18}O_2$ demonstrate that one $^{18}O$ atom from $^{18}O_2$ is incorporated into succinate (Fig. 5d). Comparison of the calculated and observed mass envelopes for $^{18}$O- and $^{16}$O-incorporation revealed 90–95% incorporation of $^{18}O$ in the products using 97% $^{18}O_2$ gas (Supplementary Fig. 21).

Incubations were performed using $^{16}O_2$ in buffer carefully made with $H_2{}^{18}O$. The results reveal formation of the hydroxylated (3R)-OH-Asp103$_{hFX}$ product with a mass of 1051.21 Da $[M + 4H]^{4+}$, showing no evidence for $^{18}O$ incorporation from $H_2{}^{18}O$ (Fig. 5e, Supplementary Fig. 20). The combined observations clearly indicate that $O_2$ is the sole source of the O atom incorporated into EGFD substrates during AspH catalysis, irrespective of whether an aspartate- or asparagine-residue undergoes hydroxylation.

## Discussion

Our results provide insight into how the 2OG oxygenase AspH binds, then catalyses hydroxylation of both Asp- and Asn-residues in EGFDs, by employing an unusual Fe(II) coordination chemistry. Structures of catalytically competent AspH:Fe(II):2OG:substrate complexes confirm that the normally conserved aspartate-/glutamate-ligand in 2OG oxygenases is replaced by a tightly bound water molecule (W1) in AspH. W1 binding is supported by second-sphere hydrogen bonds to D721 and the substrate-residue backbone carbonyl, as also observed in a catalytically inert AspH:Mn:2OG:substrate complex and modelling studies (Fig. 1)[19,39,40].

The sidechain of Q627 adopts distinct conformations to hydrogen-bond with either the substrate Asp carboxylate or Asn amide sidechain. The role of the flexible nature of AspH Q627 in enabling binding of more than one substrate is similar to the role of Q239 in FIH, another promiscuous human 2OG oxygenase[73]. Differences in W2

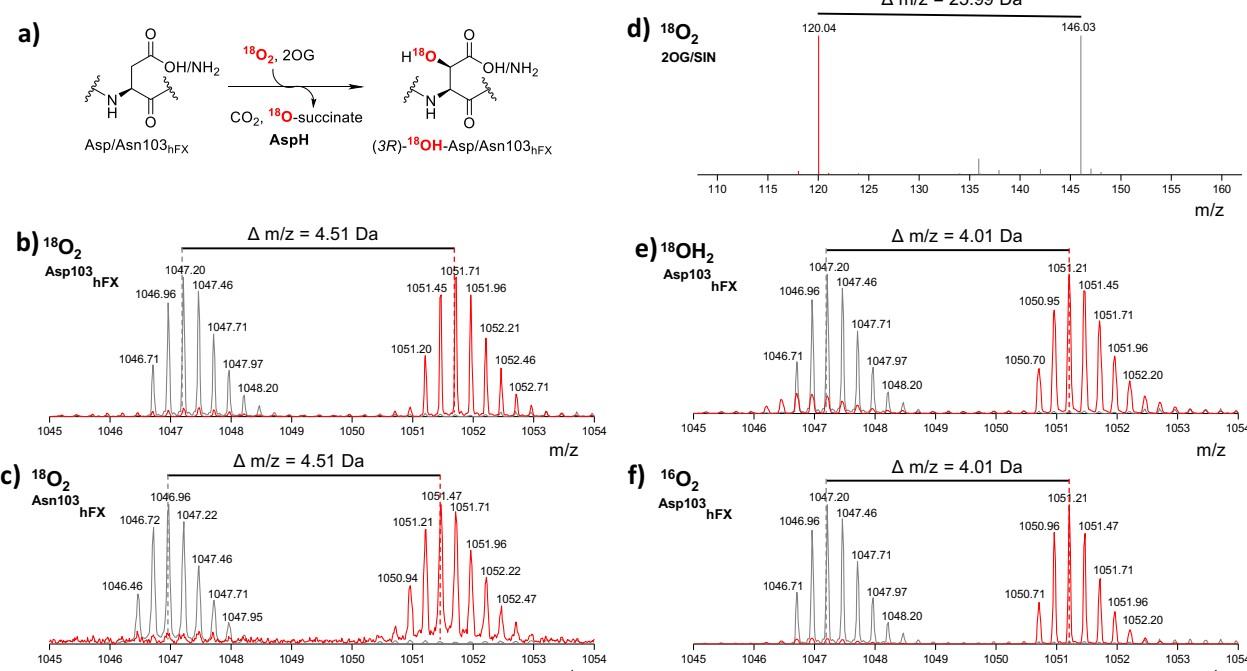

**Fig. 5 | The O atom incorporated into hydroxylated EGFD originated from molecular oxygen. a** Fate of the $O_2$ oxygen-atoms during AspH catalysis. **b, c** Mass spectrum of hydroxylated hFX-Asp and hFX-Asn after incubation with $^{18}O_2$, showing a + 4.51 Da mass shift (at charge state m/z = 4) relative to $^{16}O_2$, consistent with incorporation of $^{18}O$ from $O_2$ into the product. **d** Mass spectrum of corresponding $^{18}O_2$ incubation results in formation of $^{18}O$-labelled succinate. **e, f** No $^{18}O$ incorporation was observed when $H_2^{18}O_2$ was used, showing the product alcohol oxygen originates from $O_2$ rather than water. Labelling assays were performed by controlled exposure to $^{18}O_2$ (97% purity) of anaerobic samples ($O_2$ concentration <2 ppm) containing AspH (1 μM), FAS (20 μM), L-ascorbic acid (100 μM), hFX-Asp (20 μM) or hFX-Asn (20 μM), and 2OG (100 μM) using a Schlenk line setup, with analysis using HPLC coupled to a QTOF mass spectrometer. Grey spectra: no 2OG controls; red spectra: turnover conditions. Calculated $^{18}O$ product mass: 1051.7 Da [M + 4H]$^{4+}$; Calculated $^{16}O$ product mass: 1051.2 Da [M + 4H]$^{4+}$, m/z = mass over charge state.

binding between hFX-Asn versus hFX-Asp structures suggests that substrate binding involves a Fe−W2−W3−Q627 hydrogen-bond network, which, although not essential, has potential to modulate reaction rates in a manner relating to substrate identity, including the more efficient oxidation of hFX-Asp compared to hFX-Asn (Supplementary Fig. 4).

Studies with 2OG oxygenases using $^{18}O/H_2^{18}O$ have shown that a single O-atom is incorporated from $O_2$ into succinate[74], whereas in some cases hydroxylated products may derive oxygen from both $O_2$ and $H_2O$[63,75]. In contrast, and consistent with recently reported modelling studies[39], our $^{18}O$-labeling studies confirm that AspH incorporates $^{18}O$ exclusively from $^{18}O_2$ into both succinate and the (3R)-hydroxylated residue, despite the presence of the Fe-coordinating W1 water (Fig. 5). Studies with the catalytically inert, isoelectronic $O_2$ mimic NO, both in solution and in microcrystals (Fig. 3), show it binds Fe *trans* to H725 displacing W2, forming a high-spin {FeNO}$^7$ complex. Furthermore, photolysis−recovery EPR experiments reveal a low activation energy (~0.12 kJ/mol) for NO re-binding, reflecting a dynamic Fe-coordination sphere that may reflect efficient catalysis by AspH and its potential $O_2$-sensing role[76]. Complementary X-ray emission spectroscopy studies show that AspH returns to the Fe(II) state after a single turnover. Together, these data support a catalytic mechanism involving $O_2$ binding *trans* to H725, formation of the Fe(IV)=O intermediate, hydrogen abstraction, radical rebound, and regeneration of the Fe(II) resting state, as supported by modelling studies[39].

High-resolution product complexes demonstrate the (3R)-stereochemistry of Asp- and Asn-hydroxylation[15,67] and reveal product alcohol coordination to Fe *trans* to H725, mirroring W2's position in the substrate complexes. The absence of W3 and partial or complete exchange of succinate with 2OG in both the cryogenic and time-resolved structures suggests succinate shifts from bidentate to monodentate coordination prior to dissociation, thereby enabling 2OG re-binding and another catalytic cycle. Notably, Q627 can adopt an alternative conformation after turnover, engaging the hydroxy group of S668 and disrupting the Fe−W2−W3−Q627 network, a state which likely weakens succinate coordination and promotes its release to facilitate 2OG rebinding.

Despite an Fe(II) coordination site resembling that of the 2OG and Fe(II) dependent halogenases[36,37], AspH does not catalyse EGFD halogenation[18]. The AspH:Fe:2OG:isothiocyanate:substrate complex shows isothiocyanate binding at the W2/$O_2$ site but not at the W1 site; halides/pseudohalides have been observed to coordinate to halogenases at the W1 analogous site to enable halogen transfer[36,37,65,77]. This observation suggests that the tightly bound W1 is not suited to (pseudo)halide displacement. Previous studies have shown that the AspH H679A and H725A variants, each with only a single histidine ligand, retain catalytic activity, albeit at reduced efficiency[57], whereas the D721A variant which destabilizes W1 is substantially less active than wildtype AspH[38,39]. Together, these results suggest that W1 is important for productive hydroxylation and that its presence hinders (pseudo)halide binding at the Fe site. It is possible that the preference for two histidine-residues and an additional W1 for AspH over the typical HXD/E motif may have evolved to avoid interference with EGFD-bound Ca(II). EGFDs bind Ca(II) in close proximity to the AspH hydroxylation site, and a carboxylate ligand in AspH could enable disruptive interprotein Ca(II) contacts, possibly impeding product release. Note, however, that Ca(II) does not inhibit AspH catalysis[41]. This structural insight highlights opportunities to engineer AspH variants for halogenation by targeting the stability of W1, e.g., through

D721A or second-sphere mutations[35], or by modifying the primary coordination sphere (e.g., H679/725 substitution[57]) to promote W1 destabilization. Such engineered enzymes could not only shed light on the physiological relevance of EGFD hydroxylation, but also broaden the repertoire of biocatalysts for selective halogenation chemistry.

The relatively high AspH $K_M$ for $O_2$ indicates its potential for involvement in hypoxia sensing[76]. The dynamic AspH Fe(II) coordination sphere, involving W2, W3 and Q627, may not only fine-tune catalytic reactivity but act as a mechanism for regulating its rate of reaction with $O_2$, in a manner related to but different to that proposed for the hypoxia inducible factor-α prolyl-hydroxylases (PHDs)[34].

SFX enables reactions in crystals to be analysed under near-physiological conditions with limited radiation damage. The use of large single crystals under cryogenic conditions can restrict (co)substrate diffusion and product release due to crystal lattice constraints. Compared to traditional cryogenic crystals, microcrystals offer improved diffusion of $O_2$ and substrate, enhancing the likelihood of capturing catalytic states[78]. Our optimized microcrystal and serial crystallography setup lays the foundation for time-resolved studies on AspH using SFX, with potential to capture transient intermediates in real time. Such experiments would enable visualization of short-lived Fe−$O_2$ and Fe(IV)=O intermediates that are not captured in the present structures, providing missing insight into $O_2$ activation and hydroxylation chemistry of AspH. SFX/XES studies on IPNS, which belongs to the 2OG oxygenase structural superfamily, have enabled characterisations of an IPNS:Fe(III)-superoxide complex and revealed conformational changes during catalysis[25,52]. Future work will focus on employing this approach to stall AspH catalysis using substrate or 2OG analogues, and on applying rapid mixing to track intermediate formation and decay at high temporal resolution.

The observation that W1 is essential for productive AspH catalysis is of interest with respect to inhibitor development. AspH is a potential anticancer target, because it is overexpressed and translocated to the surface of various cancers, in a manner correlating with increased cancer cell motility and poor prognosis[10,11,14,79,80]. Small-molecule inhibition of AspH reduces cancer cell invasiveness[14], though how this observation is linked to AspH catalysis is unclear. AspH inhibitors will be of use in investigating its role in healthy biology and disease. However, all structurally characterized AspH inhibitors to date are 2OG mimics, such as pyridine-2,4-dicarboxylic derivatives, which bind in a manner that closely resembles 2OG itself[43,81,82]. Since many 2OG oxygenase inhibitors share this binding mode, selectivity remains a challenge. Therefore, developing inhibitors that target the AspH active site by displacing W1 (i.e., binding trans to the 2OG C2 carbonyl oxygen) represents a promising strategy to improve selectivity for AspH inhibition.

## Methods
### AspH production and purification
Purified ( > 95% by SDS-PAGE and MS) recombinant human AspH$_{315-758}$ was produced using *E. coli* BL21 (DE3) pLysS competent cells (Agilent Technologies) as reported[53]. In brief, AspH$_{315-758}$ was produced from a pET-28a(+) construct encoding N-terminally His$_6$-tagged AspH$_{315-758}$ (UniProt Q12797), transformed into BL21 (DE3) pLysS cells. Single colonies were used to inoculate 2×YT starter cultures containing kanamycin (50 μg·mL$^{-1}$), which were then used to seed large-scale expression cultures in autoinduction medium supplemented with kanamycin. Cultures were grown at 37 °C until an OD$_{600}$ of 0.8, then at 18 °C overnight. Cells were harvested by centrifugation ($\approx$ 11,800 × g, 4 °C) and stored at −80 °C. For purification, frozen cell pellets were resuspended in binding buffer (50 mM HEPES, pH 7.5, 500 mM NaCl, 5 mM imidazole) supplemented with EDTA-free protease inhibitor cocktail and DNase I. Cells were lysed using a high-pressure cell disruptor, and insoluble material was removed by centrifugation ($\approx$ 58,000 × g, 30 min, 4 °C). The clarified lysate was filtered and loaded onto a Ni(II)-affinity column pre-equilibrated in binding buffer. After washing with buffer containing 40 mM imidazole, His$_6$-tagged AspH$_{315-758}$ was eluted using an imidazole gradient up to 500 mM. To obtain untagged AspH$_{315-758}$, pooled Ni(II)-affinity fractions were treated with thrombin (0.11 units per mg His$_6$-tagged AspH$_{315-758}$) and dialysed overnight at 4 °C against binding buffer to remove imidazole and enable tag cleavage. The reaction mixture was applied to reverse Ni(II)-affinity chromatography, and AspH$_{315-758}$ was collected in the flow-through and wash fractions. For preparation of metal-free protein, the combined AspH$_{315-758}$ fractions were dialysed into gel filtration buffer (50 mM HEPES, pH 7.5, 150 mM NaCl) supplemented with EDTA (30 mM) and 1,10-phenanthroline (5 mM). The sample was then concentrated and further purified by size-exclusion chromatography on a Superdex 75 column (Cytiva) using Chelex-treated gel filtration buffer. Purified AspH$_{315-758}$ was concentrated (typically 50−350 μM), aliquoted, flash-frozen in liquid N$_2$, and stored at −80 °C.

### Preparation of AspH crystals for cryo-crystallography
An AspH crystallization solution was prepared by mixing AspH ( ~ 370 μM in 50 mM HEPES, pH 7.5, 150 mM NaCl), 2OG (2 mM), and ferrous ammonium sulphate x 6 H$_2$O (FAS) (1 mM), followed by the addition of hFX-Asp or hFX-Asn ( ~ 800 μM) at 4 °C. A crystallization screen was conducted by varying the pH (0.1 M bis-tris propane, pH 6.5−8.5 in 0.5 pH increments, vertical axis) and PEG concentration (PEG 3350, 18−22 in 1% increments, horizontal axis), with 0.2 M NaBr or KSCN. AspH crystals were grown using the hanging drop method by combining 1 μL of reservoir solution with 2 μL of protein solution, using 500 μL of precipitant solution in 24-well hanging drop VDX plates (Hampton Research) within an anaerobic chamber (Belle Technologies; <2 ppm O$_2$). The plate was then transferred to a fridge and maintained at 4 °C. Crystals formed within 24 to 72 h. Crystals were supplemented with cryoprotectant (mother liquor with 25% v/v PEG 400, mixed 1:1 with the crystal-containing drop), harvested using a nylon loop, and cryocooled by rapid plunging into liquid nitrogen before data collection. Diffraction data for single crystals were collected at 100 K using synchrotron radiation at the Diamond Light Source beamline I03. The data were indexed, integrated, and scaled using the Xia2 pipeline (Supplementary Table 2). Structures were solved by molecular replacement with Phaser[83], using PDB entry 8RE9[40] as the search model. Structures were iteratively refined using PHENIX (version 1.21.1.5286)[84] and manual model building using COOT (WinCOOT 0.9.8.7)[85]. For product complex structures, 2-oxoglutarate (2OG) and succinate were modelled as alternative ligands occupying the same cosubstrate binding site and refined with reciprocal occupancies constrained to a combined occupancy of 1.0, with final values selected based on best agreement with electron density, ligand and neighbouring residue B-factors, and overall refinement statistics.

### Sample preparation for SFX and SSX experiments
AspH$_{315-758}$ seed crystals were prepared in aerobic conditions using a batch method; AspH (100 μL of ~20 mg·mL$^{-1}$ AspH$_{315-758}$ in 50 mM HEPES, pH 7.5 and 150 mM NaCl) was mixed with MnCl$_2$ (2.5 μL of 100 mM in Milli-Q), 2OG (2.5 μL of 100 mM in Milli-Q), and hFX-Asp (1.2 mg), and incubated at 4 °C for 5 min prior to mixing with precipitant solution (640 μL of 0.1 M bis-tris propane pH 7.5, 0.1 M KSCN, 16% v/v PEG 3350) and single crystal seed stock solution (10 μL). The crystallisation solution was mixed carefully and pipetted into the wells (60 μL) of a 96-well plate (PS, half area, clear, microlon, med. binding, Greiner Bio-One). The plate was sealed (Polyolefin StarSeal, Starlab, UK) and incubated at 4 °C for at least 14 h. The wells were then combined and prepared as seeds using the Seed Bead Kit (Hampton Research, USA) following the manufacturer's instructions.

Anaerobic ( < 2 ppm O$_2$) microcrystallisation was performed using a batch method; AspH (500 μL of ~20 mg·mL$^{-1}$ AspH$_{315-758}$ in 50 mM HEPES pH 7.5 and 150 mM NaCl) was mixed with FAS (12.5 μL of

100 mM in Milli-Q), 2OG (12.5 μL of 100 mM in Milli-Q), and hFX-Asp (3 mg) in an anaerobic chamber (Belle Technology, UK) and incubated at 4 °C for 5 min prior to mixing with the precipitant solution (3.2 mL of 0.1 M bis-tris propane pH 7.5, 0.1 M KSCN, 16%$_{v/v}$ PEG 3350) and a seed stock solution (10 μL). The resultant crystallisation solution was mixed carefully and divided into the wells (60 μL) of a small volume 96-well plate (PS, half area, clear, med. binding plates, Greiner Bio-One). The plate was sealed (Polyolefin StarSeal, Starlab, UK) and incubated at 4 °C for at least 48 h. The microcrystal slurry was pooled, allowed to settle at 4 °C, and subsequently washed 2x with precipitant solution to remove excess Fe. Microcrystals were stored at 4 °C.

### Data collection of SFX and SSX datasets

Room temperature diffraction data for microcrystal slurries were collected at the Macromolecular Femtosecond Crystallography (MFX) beamline of LCLS, I24 of the Diamond Light Source (DLS), or at the nano-crystallography and coherent imaging (NCI) beamline of PAL-XFEL[86,87].

At the LCLS, the ADE drop-on-tape method was used to collect anaerobic diffraction data for the AspH:Fe:2OG:hFX-Asp complex[48]. Droplets were ejected onto the conveyor belt at room temperature in a He atmosphere at a flow rate of 6 μL/min. The X-ray wavelength was 1.27 Å (9.83 keV) with a data collection rate of 30 Hz, a pulse photon energy of 1.5–2 mJ, a pulse length of 35 fs and a beam size of 2.5 × 2.5 μm$^2$. X-ray diffraction data (XDR) were collected with a Rayonix MX340-HS detector. A structure of AspH:Fe:2OG:hFX-Asp that was exposed to $O_2$ for 1.5 s was obtained using the same conditions, in addition to passing the droplets through a 100% $O_2$ reaction chamber at a conveyor belt speed of 99 mm/s.

The AspH:Fe:2OG/succinate:(3R)-OH-Asp103$_{hFX}$ complex structures were obtained by exposing the anaerobically prepared crystal slurries to air. At the PAL-XFEL, fixed-target data collection was used. Samples (60 μL) were loaded on a 41 μm mesh[70], and data were collected at a step size of 50 μm. The X-ray wavelength was 1.30 Å (9.5 keV) with a data collection rate of 30 Hz, and a detector distance of 123 mm. XDR was collected with a Rayonix MX225-HS detector. At DLS, fixed-target data collection was used; samples (100 μL) were loaded onto a silicon chip[88]. The X-ray wavelength was 1.00 Å (12.4 keV) with a pulse photon energy of 12.4 keV, and a beam size of 7 μm x 7 μm. XDR was collected with a Dectris PILATUS3 6 M detector.

### X-ray emission spectroscopy

X-ray emission data were collected in tandem with diffraction data using a multicrystal wavelength-dispersive hard X-ray spectrometer in von Hamos geometry with the analyser crystals above the X-ray interaction point and the position sensitive detector at 90 degrees from the beam direction in the horizontal plane[89,90]. An array of three cylindrically bent (R = 250 mm) LiNbO$_3$ (23$\bar{4}$) analyser crystals was placed 250 mm above the interaction point with the centre of the crystals at 80.69 degrees with respect to the interaction point, collecting both Fe K$_\alpha$ lines on an ePix 100 detector with its centre located -82 mm to the side of the X-ray interaction point. Spectra from a Fe metal foil were collected for calibration. The XES data were pedestal corrected to account for differences in noise of the detector pixels and for background subtraction, a slice on either side of the region of interest (ROI) was selected. A row-by-row first order polynomial fitting scheme was utilized for the initial two-dimensional background subtraction. To bring the baseline of the spectra to zero, a one-dimensional background subtraction (also utilizing a first order polynomial fitting scheme) was also deployed. Spectra were smoothed using a Savitzky-Golay filter with a window length of 9 and polynomial order of 3 and subsequently were area normalized. Full Width Half Max (FWHM) values for the K$_{\alpha1}$ line were calculated numerically using an energy window of 6399 to 6406.5 eV. Error bars for the FWHM were calculated using a bootstrapping procedure as described[91].

### Data processing and model building of SFX datasets

Data processing was carried out as described[47]. In brief, during the SFX experiments, AspH data were monitored and processed using dials.stills_process-based scripts (in the case of PAL-XFEL data)[92] and cctbx.xfel (in the case of LCLS data)[93] using dials.stills_process to index and integrate diffraction images. The initial estimation of detector geometry (Beam centre and distance) was measured from Ag(I) behenate (in the case of LCLS data) or ceric oxide (in the case of PAL-XFEL data) powder diffraction patterns. A first round of detector metrology refinement was performed[94], followed by dials.stills_process based strong spots indexing. One or more rounds of geometry refinement were carried out until no significant increasing of the indexing rate was observed. The detector geometry was checked and refined periodically during experiments to identify any shifts of the sample position. Integrated data were scaled and merged using cctbx.xfel.merge, with a reference model that has unit cell parameters of a = 50.796 Å, b = 88.16 Å, c = 125.161 Å, α = β = γ = 90° (space group P2$_1$2$_1$2$_1$). The merging resolution was selected based on several criteria, including where the multiplicity drops below ten-fold, the CC$_{1/2}$ no longer decreases uniformly and the completeness beyond 99%.

Structures were solved by molecular replacement with Phaser[83], using PDB entry 8RE9 as the search model. Structures were iteratively refined using PHENIX (version 1.21.1.5286)[84] and manual model building using COOT (WinCOOT 0.9.8.7)[85].

B-factor analyses were performed using PyMOL (version 3.0, Schödinger, LLC). For each refined structure, main chain atoms were selected and B-factors were extracted. The B-factors were normalised using a Z-score approach, computing the mean and the standard deviation of the B-factors. If the standard deviation was zero (indicating no variance), we used 1 as a default value to avoid division by zero. B-factors were then normalised by subtracting the mean and dividing by the standard deviation. Then, the normalised B-factors were applied to the corresponding main chain atoms, and the structure was rebuilt. B-factors were visualised in a blue-white-red spectrum between -2 and 2 to highlight the variations. The same protocol was applied separately to sidechain atoms.

### Sample preparation for UV-Vis experiments and solution EPR experiments

Samples were prepared under anaerobic conditions (< 2 ppm $O_2$) in an anaerobic chamber (Belle Technology, UK). Stock solutions of UV-Vis buffer (50 mM HEPES, pH 7.5, 150 mM NaCl, 20% glycerol) and solids (FAS, 2OG, and hFX-Asp) were placed in the anaerobic chamber to remove residual $O_2$. Stock solutions of apo-AspH (~68 mg·mL$^{-1}$ AspH$_{315-758}$, 1.3 mM) were transferred into the anaerobic chamber prior to use. Solutions of 2OG (100 mM in $H_2O$) and FAS (100 mM in $H_2O$), were prepared in the anaerobic chamber. The total volume was 75 μL and contained 1 mM AspH, 4.75 mM FAS (4.75 equiv.), 5 mM hFX-Asp (5 equiv.) and residual UV-Vis buffer. Nitric oxide (NO; 1,000 ppm in $N_2$, 60 min) exposure of samples involved the use of a custom glass apparatus as described[25].

### UV-Vis experiments

UV-Vis spectra were recorded using a Carry 3500 UV-Vis Compact Peltier spectrometer (Agilent, US). Samples were prepared and data collected in a rectangular, quartz, ultra-micro, open top cuvette (2.5 × 5 mm, 10 mm pathlength, minimal volume: 70 μL), and sealed with a rubber septum in an anaerobic chamber (< 2 ppm $O_2$, Belle Technology, UK). Data were recorded at 293 K, with a scan range from 200-800 nm (1 nm data interval). The UV-Vis spectra were processed with the Carry UV workstation (Agilent, US).

### Sample preparation for microcrystal EPR experiments

Anaerobic (< 2 ppm $O_2$) microcrystallisation was performed using the batch method as described above, using either hFX-Asp or hFX-Asn,

and FeSO$_4$·7H$_2$O instead of FAS. The microcrystal slurry was pooled and allowed to settle at 4 °C, then washed 2x with precipitant buffer to remove excess Fe. The precipitant buffer (1 mL of: 0.1 M bis-tris propane, pH 7.5, 0.1 M KSCN, 16% v/v PEG 3350) was incubated for 30 min with pure NO gas ( > 99.9%) using a custom glass apparatus, as described[25], to saturate NO in the buffer. Excess precipitant solution was removed from the microcrystal slurry and NO-saturated precipitant solution was added in a 1:1 (v/v) ratio.

## EPR experiments
Samples were prepared and transferred into a 1.2 ID x 1.6 mm OD clear-fused quartz tube ("Ilmasil" quartz from Qsil GmbH), followed by cryo-cooling and storage in liquid N$_2$. X-band continuous wave (CW) EPR data were collected using a Bruker BioSpin EMXmicro spectrometer equipped with a Premium bridge, a Bruker ER4122-SHQE-W1 resonator, an Oxford Instruments ITC-503S temperature controller, and ESR900 helium flow cryostat. Photolysis was achieved with unfiltered white light LED of a Schott KL2500 source of irradiance 670 mW/cm$^2$ delivered by a liquid light guide to the 66% transmission resonator window grill. The EPR simulation and data processing were performed in the MATLAB 24.1.0 scripting environment (The MathWorks, Inc., Natick, NJ) with the simulation routines from the EasySpin package, version 6.0.0[95].

## AspH kinetics
Kinetic studies with isolated recombinant AspH$_{315-758}$ were performed using solid phase extraction coupled to mass spectrometry (SPE-MS) assays, as described[57]. In brief, AspH-catalysed hydroxylation of hFX-Asn/hFX-Asp was monitored in real time using SPE-MS. The AspH reaction was started by addition of an appropriate amount of AspH$_{315-758}$ to the substrate mixture, containing hFX-Asn/hFX-Asp, LAA, 2OG, and Fe(II), in buffer (25 mM HEPES, pH 7.5, 50 mM NaCl, 20 °C) in a 2 mL 96 deep well plate (Greiner). Final concentrations: AspH$_{315-758}$ (0.05 μM), hFX-Asn (4.0 μM), LAA (100 μM), and the shown concentrations of 2OG and FAS (Supplementary Fig. 4). Samples were purified using a C4 cartridge and analysed using a RapidFire RF 365 high-throughput sampling robot (Agilent) attached to an iFunnel Agilent 6550 accurate-mass quadrupole time-of-flight mass spectrometer operated in the positive ionization mode. The m/z + 4 charge states of hFX-Asn/hFX-Asp were used to extract ion chromatogram data; peak areas were integrated using RapidFire Integrator 4.3.0 (Agilent). Data was exported into Microsoft Excel and used to calculate the percentage conversion; data was analysed using GraphPad prism to determine turnover numbers ($k_{cat}^{app}$) and Michaelis constants ($K_m^{app}$) for 2OG and Fe(II) with hFX-Asn (Fig. 1).

## $^{18}$O$_2$ and H$_2$$^{18}$O AspH assays
Labelling experiments were conducted under controlled $^{18}$O$_2$ (97% $^{18}$O, Merck) conditions using a Schlenk line setup. Initially, all to-be-used solutions and solids were transferred into an anaerobic chamber (Belle Technology, O$_2$ concentration: <2 ppm) and equilibrated overnight. AspH$_{315-758}$ (1 μM) was mixed with FAS (final concentration: 20 μM), 2OG (20 μM, 50 μM, or 100 μM), L-ascorbic acid (100 μM) and hFX-Asp (20 μM) or hFX-Asn (20 μM) in 50 mM HEPES (pH 7.0, 7.5, 8.0, or 8.5) with 50 mM NaCl (total volume: 200 μL). The anaerobic sample was placed into a gas-tight customized 96-well plate holder with a gas inlet. This plate holder setup was removed from the anaerobic chamber and connected to the Schlenk line, which was attached to an $^{18}$O$_2$ source (97% $^{18}$O, Merck). Residual O$_2$ was purged from the system with alternating Ar and vacuum cycles. To create a mild vacuum in the plate holder setup, the pressure was adjusted to 700 mbar, and the gas inlet was opened. The system was then filled with $^{18}$O$_2$ and equilibrated for 5 min. The setup was sealed, and samples were exposed to $^{18}$O$_2$ for at least 60 min; the reaction was subsequently quenched by addition of 10% v/v formic acid. MS-analyses were performed by LC-MS using a

Xevo G2-XS (Waters) equipped with an ACE 5-AQ (Avantor), and analysed using MassLynx. For the control experiments in $^{18}$O-water, AspH was buffer exchanged into HEPES buffer (50 mM in $^{18}$O-water ( > 98 atom-% $^{18}$O), pH 7.5) using Zeba™ Spin Desalting Columns (Thermo-Fisher, UK). All cofactor stock solutions were prepared in $^{18}$O-water and then diluted to the final working concentration using HEPES (50 mM in $^{18}$O-water ( > 98 atom % $^{18}$O), pH 7.5). The assay mixtures were then prepared as described above.

Detection of 2OG and succinate in assay samples was performed by Liquid Chromatography Mass Spectrometry (LCMS) using an Agilent 1290 infinity II LC system connected to an Agilent 6550 accurate mass iFunnel quadrupole time of flight (QTOF) mass spectrometer. Assay samples were diluted 10-fold in 100% acetonitrile; 4 μL of the sample was injected and loaded onto a *InfinityLab Poroshell 120 HILIC-Z*, 2.1 ×50 mm, 2.7 μm column (Agilent). 2OG and succinate were separated using a step wise gradient (0 min–100% Solvent B, 5.0 min–100% Solvent B, 15.0 min–0% Solvent B, 20 min–0% Solvent B, 25 min–100% Solvent B, 40.0 min–100% Solvent B), using a flow rate of: 0.25 mL/min; Solvent A: LCMS grade water containing ammonium acetate (10 mM, pH 9.0), Solvent B: acetonitrile:solvent A (90:10 v/v). The mass spectrometer was operated in the negative ionisation mode with a drying gas temperature (200 °C), drying gas flow rate (13 L/min), nebulizer pressure (40 psig), sheath gas temperature (300 °C), sheath gas flow rate (12 L/min), capillary voltage (3000 V), nozzle voltage (0 V), fragmentor voltage (125 V). Acquired data were analysed using Agilent MassHunter Qualitative Analysis (Version B.07.00) software.

## Reporting summary
Further information on research design is available in the Nature Portfolio Reporting Summary linked to this article.

## Data availability
All data needed to evaluate the conclusions in the paper are present in the paper and/or the Supplementary Materials. The atomic coordinates and structure factors are deposited in the PDB accession codes: 9FVX (AspH:Fe:2OG:hFx-Asn, MX), 9FVZ (AspH:Fe:2OG:hFx-Asp, MX), 9FVW (AspH:Fe:2OG/succinate:hFx-(OH)Asn, MX), 9FVY (AspH:Fe:2OG/succinate:hFx-(OH)Asp, MX), 9FVU (AspH:Fe:2OG:hFx-Asn:SCN, MX), 9FW0 (AspH:Fe:2OG:hFx-Asp, SFX), 9HO2 (AspH:Fe:2OG:hFX-Asp, SFX), 9HO1 (AspH:Fe:2OG:hFx-(OH)Asp, 1.5 s O$_2$, SFX), 9HO0 (AspH:Fe:2OG:hFx-(OH)Asp, SFX), 9NHZ (AspH:Fe:2OG:hFx-(OH)Asp, SSX), 9FVV (AspH:Mn:2OG:hFx-Asn:SCN, MX) and 9HO3 (AspH:Fe:2OG:hFx-Asp:NO, MX). The mass spectrometry and X-ray emission spectroscopy data generated in this study are provided in the Source Data file. Source data are provided with this paper.

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

## Acknowledgements

We thank the staff at LCLS and PAL-XFEL. We also thank the staff at Diamond Light Source UK for beam time at beamline I03 and I24 (proposals MX-32727and MX-31353). A.B. acknowledges funding from the Biotechnology and Biological Sciences Research Council (BB/M011224/1). P.R. thanks the Wellcome Trust (227298/Z/23/Z). A.M.O. was supported by a Strategic Award from the Wellcome Trust (227298/Z/23/Z), the Biotechnology and Biological Sciences Research Council, a Wellcome Investigator Award (210734/Z/18/Z) and a Royal Society Wolfson Fellowship (RSWF\R2\182017). C.J.S. thanks the Ineos Oxford Institute, the Wellcome Trust (106244/Z/14/Z), the Biotechnology and Biological Sciences Research Council and Cancer Research UK for supporting work on oxygenases. A.T.S acknowledges funding from the EPSRC (EPW524311/1) and Diamond Light Source. This work was supported by the National Institutes of Health (NIH) Grants GM110501 (J.Y.) and GM126289 (J.F.K.). The DOT instrument used in this research was funded by Department of Energy (DOE), Office of Science, Office of Basic Energy Sciences (BES), Division of Chemical Sciences, Geosciences, and Biosciences (to J.F.K., J.Y., and V.K.Y.). XFEL data were collected under proposals L-10002/L-10015 and L-10155 at the LCLS, SLAC, Stanford, USA, and under proposal 2024-1st-NCI-I001 at the PAL_XFEL, South Korea. Experiment support at the LCLS was partially facilitated by the NIH grant P41GM139687. The Rayonix detector used at LCLS was supported by the NIH (S10 OD023453). Data processing was supported by the US DIALS National Resource (R24GM154040). Use of the LCLS, SLAC National Accelerator Laboratory, is supported by the U.S. DOE, Office of Science, BES, under contract no. DE-AC02-76SF00515. Data processing was performed in part at the National Energy Research Scientific Computing Center (NERSC), a user facility supported by the DOE, Office of Science, under contract no. DE-AC02-05CH11231, using NERSC award BES ERCAP0031621 (project lcls, 2025).

## Author contributions

P.R., C.J.S., L.B. and J.F.K. designed the study. A.Br., M.d.M. and P.R. produced the proteins. A.Br., M.d.M. and P.R. conducted crystallographic studies and analysed the data. M.d.M. and W.M. conducted EPR experiments and analysis with support from P.R., A.Br., M.d.M., A.T. and P.R. conducted labelling/MS experiments. Y.W. performed the kinetic assays, with support from L.B., T.Z., D.W.P., A.Bh. processed SFX data, K.C. and H.M. processed XES data. A.Br., M.d.M., S.A.M., P.S.S., P.A., A.S., D.A., H.M., D.W.P., V.T., A.T.S., S.D., H.S., D.R., R.A.-M. A.Bh., J.Y., V.K.Y., J.P., S.P., A.M.O., J.F.K., P.R. supported the serial crystallography experiments at Diamond Light Source, PAL-XFEL and LCLS. M.d.M., P.R., C.J.S., L.B. wrote the paper, with contributions from all authors.

## Competing interests

The authors declare no competing interests.

## Additional information

**Mariska de Munnik**[1,9], **Amelia Brasnett**[1,9], **Tiankun Zhou**[2,3], **William Myers** ⓘ [4], **Yicheng Wang**[1], **Kuntal Chatterjee**[5,8], **Anthony Tumber**[1], **Stephen A. Marshall** ⓘ [1], **Philipp S. Simon** ⓘ [5], **Pierre Aller** ⓘ [2,3], **Anastasiia Shilova** ⓘ [2,3], **Danny Axford** ⓘ [2], **Hiroki Makita** ⓘ [5], **Daniel W. Paley** ⓘ [5], **Vandana Tiwari**[5,6], **Alexander T. Stead** ⓘ [1], **Sebastian Dehe** ⓘ [6], **Humberto Sanchez**[6], **Daniel J. Rosenberg**[6], **Roberto Alonso-Mori** ⓘ [6], **Asmit Bhowmick** ⓘ [5], **Junko Yano** ⓘ [5], **Vittal K. Yachandra** ⓘ [5],

Jaehyun Park [7], Sehan Park[7], Allen M. Orville [2,3], Lennart Brewitz [1] ✉, Jan F. Kern [5] ✉, Christopher J. Schofield [1] ✉ & Patrick Rabe [1,2] ✉

[1]Chemistry Research Laboratory and the Ineos Oxford Institute for Antimicrobial Research, University of Oxford, Oxford, UK. [2]Diamond Light Source, Diamond House, Harwell Science and Innovation Campus, Didcot, UK. [3]Research Complex at Harwell, Harwell Science and Innovation Campus, Didcot, Oxfordshire, UK. [4]Centre for Advanced Electron Spin Resonance (CAESR), University of Oxford, South Parks Road, Oxford, UK. [5]Molecular Biophysics and Integrated Bioimaging Division, Lawrence Berkeley National Laboratory, Berkeley, CA, USA. [6]Linac Coherent Light Source, SLAC National Accelerator Laboratory, Menlo Park, CA, USA. [7]Pohang Accelerator Laboratory, Pohang University of Science and Technology, Pohang, Republic of Korea. [8]Present address: Lam Research, Tualatin, OR, USA. [9]These authors contributed equally: Mariska de Munnik, Amelia Brasnett. ✉e-mail: lennart.brewitz@ichf.edu.pl; jfkern@lbl.gov; christopher.schofield@chem.ox.ac.uk; patrick.rabe@chem.ox.ac.uk

