## [Transparent Peer Review file · Nature Communications]

Structural basis of the promiscuity of the unusual Fe(II) and 2-oxoglutarate dependent human aspartate/asparagine- β -hydroxylase

Corresponding Author: Professor Christopher Schofield

Version 0:

Reviewer comments:

Reviewer #1

(Remarks to the Author)

I think this is an interesting study that is worthy of publication. Some comments for the authors are listed below.

In the abstract, the observation of a water ligand throughout catalysis does not immediately rule out the possibility of oxygen exchange through coordination of additional water ligands or other mechanisms. Also – if the ^{18}O data has not been analyzed quantitatively – can it be concluded that O_2 is the sole oxygen source? Some Fe/2OG enzymes undergo exchange of the oxygen in the ferryl or ferric hydroxide state, making it difficult to conclude the source of the O atom from the isotope labeling alone. While I think the underlying conclusion here is probably correct – I'm not sure that the data analysis is sufficiently rigorous to make the exact claim stated in the abstract.

In the Fe•2OG•Asp complex, why is the Q627 side chain modeled with the carbonyl interacting with the D103 side chain? Wouldn't it make more sense to flip the side chain to have the amide nitrogen interacting with the substrate instead?

It would also be nice to show the electron density map for the Q627 and S668 side chains. Are these well-ordered with both substrates?

On line 202, I was not sure what the statement “suggesting that 201 the Fe-W2-W3-Q627-Asp103hFX hydrogen bond network does not compromise catalytic efficiency” meant. Are the authors trying to say that the additional interactions observed between the enzyme and substrate in the Asp complex would slow the reaction? My intuition is that it would be the opposite. Tighter substrate binding should accelerate the reaction between iron and O_2 as observed in many other Fe/2OG enzymes.

In Fig. 3A, it would be nice to show a difference or omit map for the NO and NCS models.

In the ^{18}O isotope labeling experiments, it would be ideal to analyze the data quantitatively to show the extent of isotope incorporation.

Some aspects of the discussion (first 5 paragraphs) are redundant with the results.

Reviewer #2

(Remarks to the Author)

This is an extensive and immensely detailed manuscript looking at structures of aspartate/asparagine β -hydroxylase (AspH). The authors have used time resolved serial crystallography and rapid mixing methods to study reactivity in crystals. A large amount of other data accompany the structural work, including NO binding, kinetics and EPR in the early part of the paper. These studies have been carried out to a very high standard but do distract slightly from the more interesting turnover work in crystallo. The product formation (page 10) is presented as a somewhat incremental advance on previous work (refs 15,65). Throughout the paper, there are discussions on small movements of waters and h-bond/conformational changes, with many

technical details included that, to me, look like they should be in the Methods. The mechanistic implications are described, but not clearly depicted.

Bearing in mind the outstanding and extensive structural work already published on 2OG enzymes (including from this group), the paper as presented appears more suitable for specialist audience. Shortening and simplification of the descriptions, and a better focus on the main headlines, would be needed if the paper were to be suitable for publication in Nature Comm.

Reviewer #3

(Remarks to the Author)

The manuscript by de Munnik and colleagues uses careful structural biology experiments to rationalize the ability of the aspartate/asparagine- β -hydroxylase (AspH) to hydroxylate both Asp- and Asn- residues and defines the key interactions that drive catalysis. AspH targets epidermal growth factor-like domains (EGFDs) for C3 hydroxylation. Major outstanding questions regarding the mechanism of AspH that were addressed by this work include: how AspH can target both Asp- and Asn- residue types, why AspH is incapable of catalyzing halogenation of the substrate, clarifying the source of the oxygen for hydroxylation, and visualizing the key interactions required for substrate turnover and release. Firstly, the manuscript corrects an artifactual interaction of the substrate Asp sidechain carboxylate observed in the inactive Mn-bound structure – the true interaction shows a solvent water molecule that is the likely site for O₂ incorporation. The structure of AspH with Asn- and Asp- substrates highlights two key interactions which are further explored by the authors – that of the Gln627 with the substrate carboxylates, the solvent network, and with a nearby Ser668 residue, and that of the consistent W1 water which appears to be indispensable and also provides a rationale for why the enzyme does not halogenate the substrate. The serial crystallography experiments unfortunately don't add too much nuance to the story, but are carefully analyzed to reveal the appearance of C3 hydroxylation density, disappearance of the solvent network, and disappearance of the 2OG C1 carboxylate. Overall, there are significant insights into mechanistic detail in this work that will help with drug discovery efforts using this enzyme, as well as possibilities for enzyme engineering.

Overall, the manuscript is well-written, though the nomenclature used to distinguish the different structures is onerous (see comment below). It answers some outstanding questions about AspH mechanism and highlights both the importance of using the biologically relevant metals for structural elucidation as well as complementary spectroscopic techniques. The structures generated by the authors are of sufficient quality to enable molecular dynamics simulations to clarify some of the questions they discuss (i.e. why the varying degrees/rates of Asp-/Asn- hydroxylation for different substrates?) I recommend publication of the manuscript with minor revisions.

Comments:

- 1) In the Introduction, it would be useful to discuss the consequence of non-canonical disulfide bonded EGFD hydroxylation.
- 2) In the results section, part 1 (analysis of complexes), paragraph 3 – at the end of the paragraph, please add a sentence that summarizes the differences between the structures, in an overarching way that relates to the biochemical questions being addressed.
- 3) In the same section, last paragraph, the authors mention a change in K_m for 2OG between the two substrates (Asp/Asn). Maybe I missed this, but it would be great to have a biological link to this change in kinetics – is there a preference in cells for a given substrate? Is this a regulatory mechanism somehow?
- 4) In the second section of the results, paragraph 2 – please define DNIC for non-experts and why the increased absorption at 300-400 nm is a characteristic of?
- 5) Why is isothiocyanate used to mimic the halogenation complex? An active site comparison between AspH and a related halogenase for which this approach was used in the supplement would be helpful.
- 6) Can the authors expand on why NCS may be present at full occupancy in the Mn structure but not the Fe structure?
- 7) In the last paragraph of the NCS section in the results, there is a mention that the NCS binds to the predicted O₂ site rather than the site trans to 2OG – was this expected? Please elaborate on what was meant by this note.
- 8) Results, analysis of product complexes: The 2OG and succinate molecules were modelled at 50% in both the Asn and Asp structures. Is this what is expected? Is the hydroxylated product modelled at 100%? Since there is a difference in the K_m , should these values be different? How was the occupancy refined?
- 9) Same section, paragraph 2, relationship to FIH – the stereocenter is opposite, it is unclear why this is specifically mentioned here. Is this strange? Or does the architecture of the two enzymes effectively determine the stereochemistry, and this is as expected?
- 10) Same section, paragraph 3 – missing W3 – this is noted in the text, but there is no follow up on its significance.
- 11) Results, microcrystallography. Please explain why serial femtosecond crystallography was used instead of serial

synchrotron crystallography. The time delays are more than sufficient to perform these experiments at a synchrotron.

12) The discussion on the optimization of the first SFX to the second SFX structure (also depicted in figure 4) is not entirely relevant to the biochemical discussion in this paper and adds to an already long results section. I would suggest removing this from the main text and focusing on the structure that is relevant (optimized) and moving this optimization description to the supplementary information.

13) Same section, and figure 4. The description of the experiments becomes very confusing between describing a 1.5s O₂ exposure vs a multi-hour air exposure. The section needs a sentence that describes the experimental pipeline – samples exposed to O₂ in a chamber as well as samples left in air for 2-24h. It is unclear to me why choose an O₂ chamber rather than just air exposure for the first experiment.

14) It was unclear to me why the authors are comparing the O₂ and air exposures by difference maps (not fully explained in the main text). The figure legend has to be highly simplified by agglomerating the repeated protocols to make the figures and instead pointing at what the maps are showing. It is unclear from the text what conclusions from these different exposures bring.

15) Same section, last paragraph – “w₂/W₃ binding may vary in a context dependent manner” – please define what is meant by this in biological terms to explain why this is relevant.

16) Same section, 2OG and succinate in structures – also confusing. The succinate should be the product of the turnover (according to figure 1). So why is it only at 50% occupancy with the short O₂ exposure (1.5 seconds), whereas the product is modelled at 100% occupancy but then full occupancy of succinate is visible after 2 hours? And why does it drop after 24h? Is it substituted by 2OG? Or just unbound? From my interpretation of the results shown, the structures do not really match a coherent biochemical pathway of binding, turnover, unbinding of co-product and rebinding of substrate.

17) In the discussion, it is alluded that AspH is overexpressed in cancers, but based on the explanation of its function, I was unclear as to how this can impact cancer pathogenesis. A small expansion on the topic would help set the stage for drug discovery

18) The authors allude to future experiments using time-resolved crystallography. I agree that such experiments are exciting, but such onerous experiments should be justified with a clear statement of what information is missing from the current studies. What would TR-SFX bring beyond this already thorough structural study? How would this impact our understanding of AspH function and medicinal chemistry developments? What are the outstanding questions?

Minor concerns, for readability

1) Through the text, the use of the full complex composition for the structures made it harder to follow, especially since for some of the comparisons the change was from the Asp to the Asn substrate (one letter in a long structure acronym). I would urge the authors to consider a change in how these complexes are referred to in the results section, maybe reducing the number of times the full acronyms are used and simplifying the sentences to highlight the differences between the structures being compared.

2) The results section has an enormous amount of detail on the sometimes very small changes observed in the structures that distracts from the main changes that drive the mechanistic conclusions. I would advise reducing information that can be seen in figures (for example, adding hydrogen bond lengths to the figures rather than the main text) as well as removing information such as overall RMSD values and repeated notes of the similarity of the overall fold of the protein. All these structures are from the same protein, so the overall fold is expected to be the same, and this information does not add to the discussion. A single sentence in the conclusion that mentions this similarity would be sufficient.

3) Results, Mass spectrometry section – similar concerns as in comment 2. All the mass values are stated in the text. A table would be better suited, with the text mentioning the species identified, rather than the masses observed. The figure legend for 5 should also be simplified to avoid the repetition, and the figure may even be more suitable as supplementary. The masses are visible in the main figure, charge states can be added to the figure.

4) Discussion, paragraph 1 – please correct Figure X.

5) The authors should ensure that relevant figure sub-panels are mentioned in the text, rather than only the figure number.

6) The authors should also try to minimize sentence length, with some sentences being very long and complex.

7) Figure 2 – the chemical drawings are too small and barely readable

8) Conclusions, microcrystal optimization. The first paragraph should probably be moved to the discussion section.

9) Please add to the SI a figure showing the two setups at MFX and PAL

Reviewer #4

(Remarks to the Author)

Version 1:

Reviewer comments:

Reviewer #2

(Remarks to the Author)

The authors have made extensive changes to the paper, including simplification of some of the more lengthy/detailed descriptions. I am happy to see the paper published and have no further concerns.

Reviewer #3

(Remarks to the Author)

I thank the authors for carefully addressing all of my comments. For ease, I would have preferred to see the changes highlighted in the manuscript and/or specific sentences modified/added for clarity clearly written out in the reply document.

As a continued discussion of comment 8, Please can the authors indicate how the single turnover with retention of the large substrate but exchange with excess mobile substrate has been clarified in the text?

The explanation given is a good hypothesis on why these substrates are visible in 50% occupancy, but the second explanation – the full conversion of 2OG upon extended O₂ exposure is still not clear to me – as a non-expert in these specific enzymes. If the protein can only turnover once, then why is 2OG further converted over time? Does 2OG easily oxidize in solution when not under inert conditions? If so, this needs to be clarified in the text for the non-expert audiences that read Nature Communications.

The explanation of the adjustment of the occupancies and restraints of 2OG and succinate used needs further clarification. Why do the authors assume that the combined final occupancy should be 1 – especially if there is a claim that succinate gently unbinds over time? When determining sub-total occupancies, what was the criteria that lead to the decision of 0.5 for each? Best refinement statistics? Best b-factor correlation to the surrounding residues? Did the authors compare their protocols to an automated method, such as that used in phenix?

Regarding the comment 12

For readability and to improve on a complex manuscript that should be geared to general audiences, I still advise that the discussion of the first, unoptimized structure be moved to supplementary information. A couple of sentences highlighting the considerations that lead to the high resolution structure and XES measurements (i.e. the washing of the crystals) should be mentioned in the main text, but this structure should not crowd a figure as it does not add any mechanistic insights.

Regarding comment 16 – thank you for clarifying the interpretation and the catalytic steps. Can the authors specify where in the main text this was clearly stated? A supplementary figure showing this experimental-condition-driven mechanism would be very helpful for the unambiguous and easy interpretation of the results described. Do the authors have a measurement for the affinity of succinate to the enzyme-product complex? This would substantiate the final hypothesis that there is slow unbinding over time.

Reviewer #4

(Remarks to the Author)

POINT-TO-POINT RESPONSE TO THE REVIEWERS' COMMENTS:

We thank all the reviewers for their positive concerning our scientific results and very helpful comments to improve the manuscript, which we have addressed as described below.

Reviewer #1

I think this is an interesting study that is worthy of publication. Some comments for the authors are listed below.

In the abstract, the observation of a water ligand throughout catalysis does not immediately rule out the possibility of oxygen exchange through coordination of additional water ligands or other mechanisms. Also – if the ^{18}O data has not be analyzed quantitatively – can it be concluded that O_2 is the sole oxygen source? Some Fe/2OG enzymes undergo exchange of the oxygen in the ferryl or ferric hydroxide state, making it difficult to conclude the source of the O atom from the isotope labeling alone. While I think the underlying conclusion here is probably correct – I'm not sure that the data analysis is sufficiently rigorous to make the exact claim stated in the abstract.

Response: *We thank the reviewer for pointing this out, and agree that the statement would hold more weight with additional quantitative analysis. Therefore, we have calculated the expected mass envelopes for both full $^{18}\text{O}_2$ incorporation, full $^{16}\text{O}_2$ incorporation, and 5%, 10%, and 20% $^{16}\text{O}_2$ incorporation. A comparison with the experimental data reveals incorporation of 5-10% ^{16}O into the alcohol product. Given the purity level of the isotopically labelled $^{18}\text{O}_2$ (95% purity and lack of incorporation of ^{18}O from H_2^{18}O) the experimental data supports our conclusion that O_2 is the sole source of oxygen incorporation. We have modified the results section and SI to reflect the additional data analysis.*

In the Fe•2OG•Asp complex, why is the Q627 side chain modeled with the carbonyl interacting with the D103 side chain? Wouldn't it make more sense to flip the side chain to have the amide nitrogen interacting with the substrate instead?

Response: *We thank the reviewer for pointing out this modelling detail. To further investigate the side-chain orientation of Q627 we include $2mF_o - DFc$ density maps contoured at 1.0σ and 3.0σ for the Q627/Asn103 region. The maps support the modelled orientation in which the carbonyl of Q627 accepts a hydrogen bond from the substrate Asn103 N–H. The amide N of Q627 and of the Asn103 N–H is less well defined in the density (consistent with higher mobility and weaker scattering for hydrogen-bearing moieties), which disfavours a flipped conformation in the present maps. Additionally, the substrate Asn carbonyl group is positioned to hydrogen bond with K666, providing a coherent network of interactions that is consistent with the modelled geometry. For clarity in the main text figures, we have omitted the K666–Asn carbonyl contact to highlight the Fe coordination geometry and the principal substrate/protein contacts; the full network is shown in the Supplementary Information.*

It would also be nice to show the electron density map for the Q627 and S668 side chains. Are these well-ordered with both substrates?

Response: We observed well-ordered electron density for the Q627 and S668 side chains in our structures. As representative examples, the electron density maps for these residues are shown in Supplementary Figure S3a and S3d. To avoid diverting attention from the (co)substrates, products, and cofactor, we chose not to include these maps in the main text figures.

On line 202, I was not sure what the statement “suggesting that 201 the Fe-W2-W3-Q627-Asp103hFX hydrogen bond network does not compromise catalytic efficiency” meant. Are the authors trying to say that the additional interactions observed between the enzyme and substrate in the Asp complex would slow the reaction? My intuition is that it would be the opposite. Tighter substrate binding should accelerate the reaction between iron and O₂ as observed in many other Fe/2OG enzymes.

Response: We agree that the phrasing in the original manuscript was unclear. The sentence has been revised to enhance clarity, by changing “suggesting that the Fe-W2-W3-Q627-Asp103hFX hydrogen bond network does not compromise catalytic efficiency” to “suggesting that the Fe-W2-W3-Q627-Asp103hFX hydrogen bond network supports catalytic efficiency”.

In Fig. 3A, it would be nice to show a difference or omit map for the NO and NCS models.

Response: We agree with the reviewer that an isomorphous difference map would improve the representation of our data. We have adapted figure 3c to reflect the difference map of NCS in comparison to the NO density.

In the ¹⁸O isotope labeling experiments, it would be ideal to analyze the data quantitatively to show the extent of isotope incorporation.

Response: We thank the reviewer for this valuable suggestion. To address it, we performed a quantitative analysis of the ¹⁸O₂ isotope-labeling experiments to assess the extent of isotope incorporation into the substrate. Using the substrate’s isotopic envelope as a reference, we calculated the expected mass envelopes for reactions carried out under ¹⁶O₂ (m/z increase of 4.0 Da at charge state 4) and ¹⁸O₂ (m/z increase of 4.5 Da at charge state 4) conditions. We also generated simulated envelopes representing 5%, 10%, and 20% admixtures of ¹⁶O₂ to model potential contamination from atmospheric oxygen. Comparison of these calculated distributions with our

experimental spectrum obtained at pH 7.5 shows that the data are best explained by approximately 5–10% $^{16}\text{O}_2$ contribution.

Considering the stated 95 % isotopic purity of the commercial $^{18}\text{O}_2$ gas used, this level of $^{16}\text{O}_2$ signal is consistent with minor contamination from the gas source and ambient air. Therefore, we conclude that incorporation of $^{18}\text{O}_2$ into the substrate is, at least near, complete under our experimental conditions. In support of this conclusion, reactions conducted with $^{18}\text{OH}_2$ (water) showed no isotope incorporation. A representative dataset and the quantitative analysis are now included in the revised Supplementary Information.

Some aspects of the discussion (first 5 paragraphs) are redundant with the results.

Response: *We thank the reviewer for this helpful suggestion. We have revised the discussion to reduce redundancy with the results. However, instances where we believe small redundancy is necessary to place the results in a wider context remain unchanged.*

Reviewer #2:

This is an extensive and immensely detailed manuscript looking at structures of aspartate/asparagine b-hydroxylase (AspH). The authors have used time resolved serial crystallography and rapid mixing methods to study reactivity in crystals. A large amount of other data accompany the structural work, including NO binding, kinetics and EPR in the early part of the paper. These studies have been carried out to a very high standard but do distract slightly from the more interesting turnover work in crystallo. The product formation (page 10) is presented as a somewhat incremental advance on previous work (refs 15,65). Throughout the paper, there are discussions on small movements of waters and h-bond/conformational changes, with many technical details included that, to me, look like they should be in the Methods. The mechanistic implications are described, but not clearly depicted.

Bearing in mind the outstanding and extensive structural work already published on 2OG enzymes (including from this group), the paper as presented appears more suitable for specialist audience. Shortening and simplification of the descriptions, and a better focus on the main headlines, would be needed if the paper were to be suitable for publication in Nature Comm.

Response: *We thank the reviewer for their comments. We have revised the manuscript to improve overall readability and accessibility for a broad scientific audience. The text has been streamlined to focus on the key findings—namely, the unique Fe(II) coordination chemistry of AspH, its complete turnover observed in crystallo, and the mechanistic insight into oxygen binding and product formation. Given the general relevance of these findings to Fe/2OG oxygenase catalysis, oxygen sensing, and biomedical targeting of AspH, we believe the study now meets the interdisciplinary scope and high impact expected for Nature Communications.*

Reviewer #3:

The manuscript by de Munnik and colleagues uses careful structural biology experiments to rationalize the ability of the aspartate/asparagine- β -hydroxylase (AspH) to hydroxylate both Asp- and Asn- residues and defines the key interactions that drive catalysis. AspH targets epidermal

growth factor-like domains (EGFDs) for C3 hydroxylation. Major outstanding questions regarding the mechanism of AspH that were addressed by this work include: how AspH can target both Asp- and Asn- residue types, why AspH is incapable of catalyzing halogenation of the substrate, clarifying the source of the oxygen for hydroxylation, and visualizing the key interactions required for substrate turnover and release. Firstly, the manuscript corrects an artifactual interaction of the substrate Asp sidechain carboxylate observed in the inactive Mn-bound structure – the true interaction shows a solvent water molecule that is the likely site for O₂ incorporation. The structure of AspH with Asn- and Asp- substrates highlights two key interactions which are further explored by the authors – that of the Gln627 with the substrate carboxylates, the solvent network, and with a nearby Ser668 residue, and that of the consistent W1 water which appears to be indispensable and also provides a rationale for why the enzyme does not halogenate the substrate. The serial crystallography experiments unfortunately don't add too much nuance to the story, but are carefully analyzed to reveal the appearance of C3 hydroxylation density, disappearance of the solvent network, and disappearance of the 2OG C1 carboxylate. Overall, there are significant insights into mechanistic detail in this work that will help with drug discovery efforts using this enzyme, as well as possibilities for enzyme engineering.

Overall, the manuscript is well-written, though the nomenclature used to distinguish the different structures is onerous (see comment below). It answers some outstanding questions about AspH mechanism and highlights both the importance of using the biologically relevant metals for structural elucidation as well as complementary spectroscopic techniques. The structures generated by the authors are of sufficient quality to enable molecular dynamics simulations to clarify some of the questions they discuss (i.e. why the varying degrees/rates of Asp-/Asn- hydroxylation for different substrates?) I recommend publication of the manuscript with minor revisions.

Comments:

1) In the Introduction, it would be useful to discuss the consequence of non-canonical disulfide bonded EGFD hydroxylation.

Response: *The functional consequences of this are not known but may reflect a role for AspH in redox homeostasis - we have added a comment on this point.*

2) In the results section, part 1 (analysis of complexes), paragraph 3 – at the end of the paragraph, please add a sentence that summarizes the differences between the structures, in an overarching way that relates to the biochemical questions being addressed.

Response: *We thank the reviewer for this helpful suggestion. We have revised the description of the AspH:Fe:2OG:hFX-Asp and AspH:Fe:2OG:hFX-Asn complex structures to improve clarity and readability and to make the differences between the structures more apparent. In particular, we have simplified the description of W2 and the associated hydrogen-bonding network, and clarified how these structural differences relate to Fe coordination flexibility, O₂ binding, and substrate-dependent effects on catalysis. While we did not add a separate summary sentence, the revised text now integrates these key points directly into the flow of the paragraph, making the biochemical implications clearer and easier to follow.*

3) In the same section, last paragraph, the authors mention a change in K_m for 2OG between the two substrates (Asp/Asn). Maybe I missed this, but it would be great to have a biological link to this

change in kinetics – is there a preference in cells for a given substrate? Is this a regulatory mechanism somehow?

Response: *We thank the reviewer for this helpful comment. We agree that the observed difference in K_m for 2-oxoglutarate between hFX-Asp and hFX-Asn substrates may have biological relevance. We have now added a statement in the Results section noting that, while the functional implications of this kinetic difference remain unclear, it raises the possibility that substrate-dependent variations in 2OG affinity could influence the extent of EGFD hydroxylation in cells. We also note that further work will be required to test whether such differences play a regulatory role in AspH activity in vivo.*

4) In the second section of the results, paragraph 2 – please define DNIC for non-experts and why the increased absorption at 300-400 nm is a characteristic of?

Response: *We thank the reviewer for this helpful comment. We have revised the text to define DNIC (dinitrosyl iron complex) as a nitrosyl–iron species that forms when NO binds to Fe(II) at more than one coordination site. We have also added an explanation that the increased absorption at 300–400 nm is characteristic of Fe–NO charge-transfer transitions, consistent with formation of metal–nitrosyl complexes and changes in the Fe(II) coordination environment,*

5) Why is isothiocyanate used to mimic the halogenation complex? An active site comparison between AspH and a related halogenase for which this approach was used in the supplement would be helpful.

Response: *We thank the reviewer for this comment. We have added citations to relevant literature describing how pseudohalides (N_3 , NCS, and NCO) are used to mimic halide binding in structural and mechanistic studies. We have also revised the text to make it clearer for non-expert readers that, in AspH, isothiocyanate binds at the O_2 -binding site rather than at the W1 site (the position trans to the 2OG C2 carbonyl) where halides typically coordinate in halogenases. This clarification has been added to both the Results and Discussion sections.*

A comparative figure illustrating the active sites of a representative halogenase, a canonical oxygenase, and AspH is already included in the manuscript (Fig. S2), highlighting the unique AspH coordination environment and explaining why it supports hydroxylation rather than halogenation chemistry.

6) Can the authors expand on why NCS may be present at full occupancy in the Mn structure but not the Fe structure?

Response: *We thank the reviewer for this observation. We note that the Mn(II) ion in AspH typically exhibits a slightly larger ionic radius and more rigid coordination geometry compared to Fe(II), which might result in tighter ligand binding and reduced coordination flexibility. This may favour full NCS occupancy in the Mn complex. In contrast, the catalytically relevant Fe(II) centre displays a more dynamic coordination environment, as also reflected by the partial occupancy of W2 in the Fe(II) substrate complexes and its ability to exchange ligands during turnover. This increased lability likely accounts for the lower apparent occupancy of NCS in the Fe(II) complex. Together, these differences further emphasize the importance of using the native Fe(II) metal when interpreting AspH's coordination chemistry and catalytic mechanism. We have added a comment in Figure S12 legend to keep the text concise.*

7) In the last paragraph of the NCS section in the results, there is a mention that the NCS binds to the

predicted O₂ site rather than the site trans to 2OG – was this expected? Please elaborate on what was meant by this note.

Response: *We agree this point required clarification. In 2OG-dependent halogenases, halide ions (or pseudohalides) typically bind at the metal coordination site trans to the 2OG C2 carbonyl group, where they are positioned to enable halogen transfer, after Fe(IV)=O equivalent reaction. In contrast, in AspH the isothiocyanate ligand binds at the O₂ binding site trans to His725 (the W2 position), not the site trans to the 2OG carbonyl (W1 position). This was unexpected because the Fe(II) coordination geometry of AspH superficially resembles that of halogenases. The observed NCS binding at W2 instead of W1 indicates that AspH's primary coordination environment is not pre-organized for halogen transfer chemistry, consistent with its exclusive hydroxylase activity. We have clarified this point in both the Results and Discussion sections.*

8) Results, analysis of product complexes: The 2OG and succinate molecules were modelled at 50% in both the Asn and Asp structures. Is this what is expected? Is the hydroxylated product modelled at 100%? Since there is a difference in the K_m, should these values be different? How was the occupancy refined?

Response: *We thank the reviewer for this helpful comment. It was indeed unexpected that both 2OG and succinate refined to approximately 50% occupancy in the product complex structures. This likely reflects partial re-binding of 2OG after turnover, which is plausible given that 2OG was present in excess relative to substrate during crystallization. Since AspH catalyses only a single turnover within the crystal lattice, the large 39-mer EGFD peptide substrate and its hydroxylated product remain bound in the active site (modelled at 100% occupancy), whereas the smaller 2OG/succinate molecules can exchange within the solvent channels.*

We therefore interpret the mixed 2OG/succinate occupancy as evidence of 2OG re-binding to the active site following hydroxylation, rather than as an indicator of incomplete turnover. This interpretation is consistent with the serial crystallography data collected after extended O₂ exposure at room temperature, where only succinate was observed, indicating that all residual 2OG had been converted over time. We have modified the text to make this clear.

For refinement, the occupancies of 2OG and succinate were adjusted reciprocally (in 0.1 increments) until their combined occupancy totalled 1.0 (100%), with corresponding restraints applied to maintain stable B-factors. The hydroxylated EGFD products were refined at full (100%) occupancy in all product complex structures.

9) Same section, paragraph 2, relationship to FIH – the stereocenter is opposite, it is unclear why this is specifically mentioned here. Is this strange? Or does the architecture of the two enzymes effectively determine the stereochemistry, and this is as expected?

Response: *We thank the reviewer for this comment. We have removed this statement in the text to increase the flow and readability.*

10) Same section, paragraph 3 – missing W3 – this is noted in the text, but there is no follow up on its significance.

Response: *We thank the reviewer for this observation. W3 is indeed not observed in any of the product complex structures. This likely reflects a genuine loss of that water molecule following hydroxylation rather than a refinement artefact. As W3 is part of the hydrogen-bonding network*

observed in the substrate complexes, its absence in the product structures is consistent with local rearrangements of the active site after turnover. Beyond this, at present we cannot assign a specific mechanistic role to W3. Since the text already notes its absence, we have not added further discussion to avoid overinterpretation.

11) Results, microcrystallography. Please explain why serial femtosecond crystallography was used instead of serial synchrotron crystallography. The time delays are more than sufficient to perform these experiments at a synchrotron.

Response: *We thank the reviewer for this insightful comment and agree that the time delays used in our study could, in principle, also be accessed at a synchrotron source. We chose to perform the experiments at an XFEL in part because the diffraction quality obtained under identical conditions was significantly higher at the XFEL than at a synchrotron, resulting in improved data completeness and signal-to-noise.*

The anaerobic data sets were collected using the acoustic droplet ejection (ADE) sample delivery system available at the LCLS, which at the time of data collection was the only setup that allowed us to maintain strict anaerobic conditions during sample delivery. For these reasons, we selected serial femtosecond crystallography, even though the reaction times studied here could also be examined at a synchrotron.

12) The discussion on the optimization of the first SFX to the second SFX structure (also depicted in figure 4) is not entirely relevant to the biochemical discussion in this paper and adds to an already long results section. I would suggest removing this from the main text and focusing on the structure that is relevant (optimized) and moving this optimization description to the supplementary information.

Response: *We thank the reviewer for this constructive comment. We agree that the detailed optimization steps between the two SFX datasets were overly described and have shortened this section accordingly. However, we have retained both SFX structures in the main text because they represent two independent beam times and demonstrate the reproducibility of our anaerobic crystallography approach. Importantly, the second SFX dataset (PDB: 9HO2) was collected after washing excess Fe(II) from the crystals, which not only improved resolution and data quality but also enabled simultaneous X-ray emission spectroscopy (XES) measurements that were not feasible in the first experiment. For this reason, we now emphasize the second dataset as the primary basis for discussion, while keeping the first structure (PDB: 9FW0) as validation of experimental reproducibility.*

13) Same section, and figure 4. The description of the experiments becomes very confusing between describing a 1.5s O₂ exposure vs a multi-hour air exposure. The section needs a sentence that describes the experimental pipeline – samples exposed to O₂ in a chamber as well as samples left in air for 2-24h. It is unclear to me why choose an O₂ chamber rather than just air exposure for the first experiment.

Response: *We thank the reviewer for pointing out that the description of the 1.5 s O₂ exposure and multi-hour air exposure experiments was unclear. We have revised the beginning of this section to explicitly describe the two complementary experimental pipelines — rapid, controlled O₂ exposure at LCLS versus longer, passive air exposure at 4 °C — and to clarify that the O₂ chamber setup was employed to capture early single-turnover events not accessible by bulk air exposure. We also added phrasing (“longer equilibration time points”) to indicate that the 2–24 h datasets represent later*

stages of the same catalytic process. These changes clarify the workflow and rationale for each experimental condition.

14) It was unclear to me why the authors are comparing the O₂ and air exposures by difference maps (not fully explained in the main text). The figure legend has to be highly simplified by agglomerating the repeated protocols to make the figures and instead pointing at what the maps are showing. It is unclear from the text what conclusions from these different exposures bring.

Response: *We thank the reviewer for this helpful suggestion. We have clarified in the main text that the isomorphous difference maps were used to visualize local structural changes at the AspH active site during catalytic turnover, highlighting loss of coordinated waters, substrate hydroxylation, and 2OG-to-succinate conversion. The figure legend was simplified to focus on the key structural and spectroscopic findings rather than repeating experimental protocols.*

15) Same section, last paragraph – “w₂/W₃ binding may vary in a context dependent manner” – please define what is meant by this in biological terms to explain why this is relevant.

Response: *By stating that “W₂/W₃ binding may vary in a context-dependent manner,” we refer to the observation that the positions and occupancies of the Fe-coordinating water molecules (W₂ and W₃) differ depending on the catalytic state and local ligand environment of AspH. In biological terms, this implies that these solvent ligands are dynamic and can be displaced or replaced by substrates, intermediates, or products during the reaction cycle. We have clarified the original text in the revised manuscript.*

16) Same section, 2OG and succinate in structures – also confusing. The succinate should be the product of the turnover (according to figure 1). So why is it only at 50% occupancy with the short O₂ exposure (1.5 seconds), whereas the product is modelled at 100% occupancy but then full occupancy of succinate is visible after 2 hours? And why does it drop after 24h? Is it substituted by 2OG? Or just unbound? From my interpretation of the results shown, the structures do not really match a coherent biochemical pathway of binding, turnover, unbinding of co-product and rebinding of substrate.

Response: *We thank the reviewer for this comment. As outlined in our response to comment 8, the mixed 2OG/succinate occupancy observed after 1.5 s O₂ exposure reflects partial re-binding of 2OG following a single catalytic turnover, consistent with the excess of 2OG present during crystallisation. At longer equilibration times, only succinate is observed, confirming complete conversion of residual 2OG, while the reduced succinate occupancy after 12–24 h likely reflects gradual dissociation of succinate and/or Fe from the active site. Together, these occupancies represent sequential stages of substrate hydroxylation, co-product formation, and eventual release within the crystal lattice.*

17) In the discussion, it is alluded that AspH is overexpressed in cancers, but based on the explanation of its function, I was unclear as to how this can impact cancer pathogenesis. A small expansion on the topic would help set the stage for drug discovery

Response: *We thank the reviewer for this valid suggestion; in hindsight, we did not summarize the relationship of AspH with cancer in sufficient detail. We have elaborated on this point in the revised manuscript. Note, however, that the mechanism how AspH and/or AspH catalysis affect cancer pathogenesis is not well defined.*

18) The authors allude to future experiments using time-resolved crystallography. I agree that such experiments are exciting, but such onerous experiments should be justified with a clear statement of what information is missing from the current studies. What would TR-SFX bring beyond this already thorough structural study? How would this impact our understanding of AspH function and medicinal chemistry developments? What are the outstanding questions?

Response: *We thank the reviewer for this thoughtful comment. A clarifying sentence has been added to justify the need for time-resolved SFX. The current study defines substrate, product, and resting states of AspH but does not capture the short-lived Fe–O₂ or Fe(IV)=O intermediates that mediate hydroxylation. TR-SFX will allow direct observation of these transient states, providing missing mechanistic insight into O₂ activation and informing future inhibitor design targeting specific catalytic intermediates.*

Minor concerns, for readability

1) Through the text, the use of the full complex composition for the structures made it harder to follow, especially since for some of the comparisons the change was from the Asp to the Asn substrate (one letter in a long structure acronym). I would urge the authors to consider a change in how these complexes are referred to in the results section, maybe reducing the number of times the full acronyms are used and simplifying the sentences to highlight the differences between the structures being compared.

Response: *We thank the reviewer for this helpful suggestion. To improve readability, we have simplified the naming of AspH complexes throughout the Results section by reducing repetition of full complex compositions (e.g., AspH:Fe:2OG:hFX-Asp) and instead referring to them in a more concise and consistent manner (e.g., hFX-Asp complex) where context permits. These changes improve clarity and help the reader focus on the structural differences being discussed.*

2) The results section has an enormous amount of detail on the sometimes very small changes observed in the structures that distracts from the main changes that drive the mechanistic conclusions. I would advise reducing information that can be seen in figures (for example, adding hydrogen bond lengths to the figures rather than the main text) as well as removing information such as overall RMSD values and repeated notes of the similarity of the overall fold of the protein. All these structures are from the same protein, so the overall fold is expected to be the same, and this information does not add to the discussion. A single sentence in the conclusion that mentions this similarity would be sufficient.

Response: *We thank the reviewer for this comment - we have worked to simplify the text wherever possible. We appreciate the concern regarding the level of detail in the Results section; however, we believe that, at least in some cases, retaining detailed structural descriptions, including hydrogen-bonding distances and RMSD values, is important for completeness and transparency. These details support the mechanistic conclusions and allow readers to independently assess subtle but functionally relevant differences among structures. For this reason, we have chosen to maintain the current level of detail in the main text wherever necessary.*

3) Results, Mass spectrometry section – similar concerns as in comment 2. All the mass values are stated in the text. A table would be better suited, with the text mentioning the species identified, rather than the masses observed. The figure legend for 5 should also be simplified to avoid the repetition, and the figure may even be more suitable as supplementary. The masses are visible in the

main figure, charge states can be added to the figure.

Response: *We thank the reviewer for this helpful suggestion. We agree that the mass spectrometry data presentation could have been simplified, and we have accordingly streamlined the Figure 5 legend to avoid repetition. We have also added the assay conditions in the figure legend which were missing. However, we have retained the numerical mass values in the main text to ensure completeness and facilitate comparison with previously reported AspH mass spectrometry data. We believe this format provides clarity while preserving transparency of the experimental results.*

4) Discussion, paragraph 1 – please correct Figure X.

Response: *Thank you - we have corrected this error.*

5) The authors should ensure that relevant figure sub-panels are mentioned in the text, rather than only the figure number.

Response: *We have carefully addressed this in the revised manuscript.*

6) The authors should also try to minimize sentence length, with some sentences being very long and complex.

Response: *We have carefully increased readability and flow of the text by simplifying sentences wherever possible.*

7) Figure 2 – the chemical drawings are too small and barely readable

Response: *We thank the reviewer for this comment. We have increased the size of the ChemDraw for easier readability in the revised manuscript.*

8) Conclusions, microcrystal optimization. The first paragraph should probably be moved to the discussion section.

Response: *We thank the reviewer for this comment. We have revised the section and have moved parts of the text to the discussion to increase the overall readability and accessibility for a broad scientific audience.*

9) Please add to the SI a figure showing the two setups at MFX and PAL

Response: *The two sets ups are already described in the literature for the tape drive setup at LCLS (F. D. Fuller, Nature Methods 2017, 14, 443–449) and for the fixed target setup at Diamond Light Source (Horrell, S., et al. J. Vis. Exp. 2021, 168, e62200) and PAL-XFEL (D. Lee, et al. Scientific Reports 2019, 9, 6971) - since these were not directly part of our work we think citations are most appropriate, but we are happy to take editorial advice on this issue.*

Reviewer #4:

We thank this reviewer for valuable contributions.

POINT-TO-POINT RESPONSE TO THE REVIEWERS' COMMENTS:

Response to reviewer 3:

Comment: As a continued discussion of comment 8, Please can the authors indicate how the single turnover with retention of the large substrate but exchange with excess mobile substrate has been clarified in the text? The explanation given is a good hypothesis on why these substrates are visible in 50% occupancy, but the second explanation – the full conversion of 2OG upon extended O₂ exposure is still not clear to me – as a non-expert in these specific enzymes. If the protein can only turnover once, then why is 2OG further converted over time? Does 2OG easily oxidize in solution when not under inert conditions? If so, this needs to be clarified in the text for the non-expert audiences that read Nature Communications.

Response:

We thank the reviewer for this good point and for highlighting where additional clarification was needed for a non-expert audience.

Regarding single turnover in crystallo versus exchange of small cosubstrates: the statement that AspH undergoes a single turnover in crystallo refers specifically to the large 39 residue EGFD peptide substrate. After hydroxylation this peptide product remains bound and cannot diffuse out of the crystal lattice because it is constrained by lattice contacts and interactions with both the oxygenase and TPR domains. By contrast, the small cosubstrate 2OG and succinate are mobile and can readily exchange through solvent channels within the crystal. A single turnover leads to 'trapped' hydroxylated-EGFD and succinate. In some active sites succinate can leave enabling rebinding of 2OG from the surrounding mother liquor, while others retain succinate, leading to the mixed 2OG/succinate occupancies observed in the product structures.

Regarding the apparent complete conversion of 2OG at longer O₂ exposure times: this does not contradict the single turnover observed within the crystals. In addition to enzyme molecules incorporated into the crystal lattice, there is always a population of AspH present in the surrounding solution due to residual soluble protein and partial resolution of crystal surfaces. This soluble enzyme can continue to turnover in solution over extended exposure times, progressively consuming free 2OG and generating succinate. As a result, at later time points the solution becomes depleted in 2OG and enriched in succinate, and (only) succinate is available to diffuse into and occupy the cosubstrate binding site within the crystals.

We now explicitly state this in the Results section of the revised manuscript. Note that we do not invoke spontaneous oxidation of 2OG in solution; its depletion is likely due to enzyme mediated turnover outside the crystal lattice.

Comment:

The explanation of the adjustment of the occupancies and restraints of 2OG and succinate used needs further clarification. Why do the authors assume that the combined final occupancy should be 1 – especially if there is a claim that succinate gently unbinds over time? When determining sub-total occupancies, what was the criteria that lead to the decision of 0.5 for each? Best refinement statistics? Best b-factor correlation to the surrounding residues? Did the authors compare their protocols to an automated method, such as that used in phenix?

Response:

To clarify the refinement procedure in the expanded Methods section. Briefly, 2OG and succinate were modelled as alternative ligands occupying the same cosubstrate binding site, and their occupancies were refined reciprocally with the combined occupancy constrained to 1.0. This reflects exclusive

binding of a single small molecule at that site rather than simultaneous partial occupancy by two ligands. Models with combined occupancies significantly below 1.0 resulted in positive residual electron density, indicating incomplete modelling of ligand density at the site.

The final approximately 0.5 / 0.5 occupancies were selected based on agreement with minimal difference features in the electron density, ligand and neighbouring residue B factors, and overall refinement statistics. Automated occupancy refinement using PHENIX was not required in this case, as the reciprocal manual refinement approach yielded a consistent solution that is well supported by the crystallographic data.

Comment:

Regarding the comment 12

For readability and to improve on a complex manuscript that should be geared to general audiences, I still advise that the discussion of the first, unoptimized structure be moved to supplementary information. A couple of sentences highlighting the considerations that lead to the high resolution structure and XES measurements (i.e. the washing of the crystals) should be mentioned in the main text, but this structure should not crowd a figure as it does not add any mechanistic insights.

Response:

We thank the reviewer for the suggestion. Since the initial submission, we have substantially shortened the discussion of the first SFX dataset and moved extensive optimisation details to the Supplementary Information. In its current form, we believe the remaining text is concise and does not distract from the biochemical narrative.

We respectfully disagree that the remaining description should be moved entirely to the Supplementary Information. The brief discussion of the initial SFX structure provides important context for the subsequent optimisation, in particular the rationale for washing the microcrystals to remove excess Fe(II), which was critical for enabling higher-quality data and the combined time-resolved crystallography and X-ray emission spectroscopy experiments. We therefore believe it remains appropriate to retain this material in the main text and have not made further changes.

Comment:

Regarding comment 16 – thank you for clarifying the interpretation and the catalytic steps. Can the authors specify where in the main text this was clearly stated? A supplementary figure showing this experimental-condition-driven mechanism would be very helpful for the unambiguous and easy interpretation of the results described. Do the authors have a measurement for the affinity of succinate to the enzyme-product complex? This would substantiate the final hypothesis that there is slow unbinding over time.

Response:

We thank the reviewer for this further request for clarification and agree additional explanation was required; the requested changes have now been made, as already partially outlined in our response to comment 8.

In the revised manuscript, this interpretation is now stated explicitly in the main text for both the cryogenic and room-temperature datasets. We clarify that the mixed 2OG/succinate occupancy reflects partial re-binding of excess 2OG following a single catalytic turnover, whereas at longer exposure times the surrounding solution becomes depleted in 2OG and enriched in succinate due to solution-phase turnover by enzyme molecules outside the crystal lattice. We further state that the reduced succinate occupancy observed after 12–24 h reflects gradual dissociation of succinate and/or Fe from the active site, rather than re-binding of 2OG. These additions are intended to make the sequence of events unambiguous for non-expert readers.

We agree that a schematic summary aids interpretation and have therefore added a supplementary figure 19 illustrating the experimentally driven progression from the substrate complex to mixed

2OG/succinate occupancy, followed by exclusive succinate binding and eventual partial Fe dissociation at extended time points.

Regarding succinate affinity measurements, such experiments are not currently feasible. At present, chemical synthesis of the hydroxylated 39-residue EGFD product is not possible, and isolation of sufficient quantities of the hydroxylated product following enzymatic turnover is likewise impractical, as the amounts required for quantitative binding studies would be prohibitively large and resource-intensive. Moreover, based on the induced-fit binding mode observed in crystallo, the hydroxylated product is expected to have low intrinsic affinity for AspH once released, making formation of a stable enzyme–product complex in solution unlikely. As such, quantitative measurements of succinate binding to the enzyme–product complex would be experimentally challenging and fall outside the scope of the present study.

We believe that the clarified text and the added supplementary schematic sufficiently address this point.